**What are the greenhouse gas observing system requirements for reducing fundamental biogeochemical process uncertainty? Amazon wetland CH$_4$ emissions as a case study.**

A. Anthony Bloom[1], Thomas Lauvaux[1,2], John Worden[1], Vineet Yadav[1], Riley Duren[1], Stanley Sander[1], David Schimel[1].

[1]Jet Propulsion Laboratory, California Institute of Technology, Pasadena, CA

[2] Department of Meteorology, The Pennsylvania State University, University Park, PA

*Correspondence to*: A. Anthony Bloom (abloom@jpl.nasa.gov)

**Abstract.** Understanding the processes controlling terrestrial carbon fluxes is one of the grand challenges of climate science. Carbon cycle process controls are readily studied at local scales, but integrating local knowledge across extremely heterogeneous biota, landforms and climate space has proven to be extraordinarily challenging. Consequently, top-down or integral flux constraints at process-relevant scales are essential to reducing process uncertainty. Future satellite-based estimates of greenhouse gas fluxes – such as CO$_2$ and CH$_4$ – could potentially provide the constraints needed to resolve biogeochemical process controls at the required scales. Our analysis is focused on Amazon wetland CH$_4$ emissions, which amount to a scientifically crucial and methodologically challenging case study. We quantitatively derive the observing system requirements for testing wetland CH$_4$ emission hypotheses at a process-relevant scale. To distinguish between hypothesized hydrological and carbon controls on Amazon wetland CH$_4$ production, a satellite mission will need to resolve monthly CH$_4$ fluxes at a ~333km resolution and with a ≤10 mg CH$_4$ m$^{-2}$ d$^{-1}$ flux precision. We simulate a range of low-earth orbit (LEO) and geostationary orbit (GEO) CH$_4$ observing system configurations to evaluate the ability of these approaches to meet the CH$_4$ flux requirements. Conventional LEO and GEO missions resolve monthly ~333km Amazon wetland fluxes at a 17.0 mg CH$_4$ m$^{-2}$ d$^{-1}$ and 2.7 mg CH$_4$ m$^{-2}$ d$^{-1}$ median uncertainty level. Improving LEO CH$_4$ measurement precision by $\sqrt{2}$ would only reduce the median CH$_4$ flux uncertainty to 11.9 mg CH$_4$ m$^{-2}$ d$^{-1}$. A GEO mission with targeted observing capability could resolve fluxes at a 2.0 – 2.4 mg CH$_4$ m$^{-2}$ d$^{-1}$ median precision by increasing the observation density in high cloud-cover regions at the expense of other parts of the domain. We find that residual CH$_4$ concentration biases can

potentially reduce the ~5-fold flux $CH_4$ precision advantage of a GEO mission to a ~2-fold advantage (relative to a LEO mission). For residual $CH_4$ bias correlation lengths of 100km, the GEO can nonetheless meet the $\leq 10$ mg $CH_4$ m$^{-2}$ d$^{-1}$ requirements for systematic biases $\leq$10ppb. Our study demonstrates that process-driven greenhouse gas observing system simulations can enhance conventional uncertainty reduction assessments by quantifying the OS characteristics required for testing biogeochemical process hypotheses.

## 1. Introduction

Quantitative knowledge of biogeochemical processes regulating global carbon-climate feedbacks remains highly uncertain (Friedlingstein et al., 2013).  Quantifying the sensitivity of biogeochemistry to climate variables directly from observations of atmospheric concentrations has long been a goal of researchers (Bacastow et al., 1980; Vukicevic et al, 1997; Gurney et al., 2008). Estimating the climate sensitivity of carbon fluxes is complicated by both the spatial scale and structure of climate anomalies and the variations of factors affecting ecosystem responses: soils, vegetation, land use and natural disturbance (King et al., 2015). Current ground-based and even space-based carbon cycle observing systems produce flux estimates at continental or even zonal resolution, limiting direct estimation of relationships between climate forcing, ecosystem properties and carbon fluxes (Huntzinger et al., 2012, Peylin et al., 2013).  The uncertainty of carbon fluxes at continental and finer scales is high, and different systems for flux estimation often produce strikingly different spatial patterns (Schimel et al 2015a; Bloom et al., 2016).  Because of the high uncertainty in the spatial regionalization of fluxes, some of the most compelling studies of carbon and climate have eliminated the spatial information and instead have used correlative approaches to identify the regions likely to be responsible for observed global concentration anomalies (Braswell et al., 1997; Cox et al., 2013; Chen et al., 2015; Franklin et al., 2016).

The expansion of surface and aircraft observing networks has increased our understanding of the carbon cycle, and is essential for precise quantification of trace gas concentrations (Andrews et al., 2014, Sweeney et al., 2015; Wilson et al., 2016). Surface networks are intrinsically limited in their density, by cost, access to remote terrestrial and marine environments, environmental conditions and other logistical constraints (Schimel et al., 2015b).  The first-generation trace

gas observing satellites were designed to make global-scale measurements of concentrations with unprecedented frequency and accuracy, but were not designed to test specific hypotheses about biogeochemical processes. The successes of GOSAT (Yokota et al., 2009) and OCO-2 (Crisp et al., 2004) open the door to designing a next generation of spaceborne greenhouse gas measurements to test specific hypotheses about the terrestrial biosphere or the oceans. In this paper, we report an observing system design exercise aimed at identifying the observing system needed to increase understanding of a long-standing uncertainty in the global carbon budget, specifically the role of tropical wetlands in the global $CH_4$ budget (Mitsch et al., 2010; Bloom et al., 2010; Melton et al., 2013). While we focus this analysis on $CH_4$, we note that the models and methodology are equally applicable to other gases (such as $CO_2$), and other regions or mechanisms.

## *Wetland CH$_4$ emissions*

Biogenic methane ($CH_4$) emission processes are one of the principal components of global carbon-climate interactions; $CH_4$ is a potent greenhouse gas (Myhre et al., 2013) and wetlands account for roughly 20-40% of the global $CH_4$ source (Kirschke et al., 2013). The processes controlling the magnitude and temporal evolution of $CH_4$ outgassing from wetland environments remain largely un-quantified on continental scales. As a result, global scale wetland $CH_4$ emissions (Melton et al., 2013) and their role in the inter-annual growth of atmospheric $CH_4$ remain highly uncertain.

Global wetland $CH_4$ emissions largely depend on soil inundation, temperature and substrate carbon availability. The major sources of wetland $CH_4$ emissions include boreal North America, boreal Eurasia, the Indonesian archipelago, the Congo and Amazon river basins (Figure 1, map) which are all characterized by high soil carbon content (Hiederer and Köchy, 2011) and substantial seasonal or year-round inundation extent (Prigent et al., 2012). By and large, Amazon wetland $CH_4$ emissions dominate both the magnitude and uncertainty of global wetland $CH_4$ emissions (Melton et al., 2013). Estimates of Amazon wetland $CH_4$ emissions range between $20 - 60$ Tg $CH_4$ yr$^{-1}$ (Fung et al., 1991; Riley et al., 2011; Bloom et al., 2012; Melack et al., 2004), roughly equivalent to $10 - 30\%$ of the global wetland $CH_4$ source. Major uncertainties are also associated with the spatial and temporal variability of $CH_4$ emissions (Figure 1). Uncertainties in tropical wetland $CH_4$ emission estimates

largely stem from a lack of quantitative knowledge of process controls on wetland $CH_4$ emissions, and a lack of data constraints on the drivers of wetland emissions. In terms of processes, a range of factors including soil pH, wetland vegetation cover, wetland depth, salinity and air-water gas exchange dynamics, likely impose fundamental controls on the rate of wetland $CH_4$ emissions. On a continental scale, spatially-explicit knowledge of carbon cycling and inundation remain

highly uncertain in the wet tropics, primarily due to a sparse in-situ measurement network, high cloud cover and biomass density

### *Top-down $CH_4$ flux estimates*

Top-down constraints on $CH_4$ fluxes – from atmospheric $CH_4$ observations – are key to retrieving quantitative information on continental-scale $CH_4$ biogeochemistry (Bousquet et al., 2011; Pison et al., 2013; Basso et al., 2016; Wilson et al., 2016). Low-earth orbit satellite missions, including SCIAMACHY, IASI, TES, and GOSAT have surveyed global $CH_4$ concentrations for over a decade (Frankenberg et al., 2008; Crevoisier et al., 2009; Butz et al., 2011; Worden et al., 2012). In particular, column $CH_4$ retrievals from SCIAMACHY have proven sensitive to wetland and other $CH_4$ emissions (Bloom et

al., 2010; Bergamaschi et al., 2013). However, cloud cover is a major inhibiting factor when measuring atmospheric greenhouse gas concentrations within the proximity of tropical wetland regions. In particular, densely vegetated seasonally inundated areas of the Amazon and Congo river basins can experience more than 95% monthly mean cloud cover. With fewer cloud-free observations of lower tropospheric $CH_4$ concentrations, atmospheric inversion estimates of wetland $CH_4$ emissions remain exceedingly difficult, especially in the absence of well-characterized prior information on the magnitude,

location and timing of emissions.

Atmospheric inverse estimates of $CH_4$ emissions are expected to improve with tropospheric $CH_4$ measurements from the upcoming ESA TROPOMI mission (Butz et al., 2012; Veefkind et al., 2012). Furthermore, geostationary missions (such as GEOCAPE) will potentially provide the measurements needed to substantially improve $CH_4$ emission estimates (Wecht et

al., 2014; Bousserez et al., 2015). Ultimately, the precision and sampling configuration of atmospheric $CH_4$ observations

both determine the observing system (OS) capability of retrieving surface $CH_4$ fluxes. It is currently unclear whether future $CH_4$ measurements will be sufficient to resolve key $CH_4$ fluxes – such as the Amazon basin wetlands – at a process-relevant resolution.

In this study we characterize the satellite observations required to quantify the biogeochemical process controls on Amazon wetland $CH_4$ emissions. Specifically, we identify and characterize the Amazon $CH_4$ emission processes (section 2.1), define the process-relevant $CH_4$ flux resolution and precision required to statistically distinguish between hypothesized wetland $CH_4$ emission scenario based on several hydrological and carbon datasets (section 2.2), we simulate atmospheric measurements throughout the Amazon basin for a range of low-earth orbit and geo-stationary orbit satellite OS, and we

derive the corresponding $CH_4$ flux uncertainty using an idealized atmospheric inversion (section 2.3). Based on our results, we establish the OS requirements and discuss the potential of future OS to resolve Amazon wetland $CH_4$ emission processes (section 3). We conclude our paper in section 4.

## 2. Methods

We construct an Observing System Simulation Experiment (OSSE) dedicated to characterizing the spaceborne OS needed to resolve the processes controlling wetland $CH_4$ fluxes from Amazon basin (Figure 2). Our OSSE involves the following 3 steps: we (1) characterize the variability of wetland $CH_4$ process controls; (2) define $CH_4$ flux resolution and precision requirements; and (3) derive the atmospheric $CH_4$ concentration OS requirements. We define the atmospheric $CH_4$ OS

requirement as the ability to meet the $CH_4$ flux resolution and precision requirements during the cloudiest time of year. We focus our analysis on March 2007: all temporally-resolved carbon and hydrological observations chosen for this study overlap in 2007, and March 2007 mean cloud cover (84%) amounts to the highest cloud cover across the whole Amazon river basin within the January – April 2007 wet season (cloud cover range = 76% - 84%) and is considerably higher than the June – September 2007 dry season cloud cover (46% - 56%).

## 2.1 Wetland process controls

Wetland $CH_4$ emissions are controlled by a range of biogeochemical processes: inundation is likely to be a first order control of wetland emissions, as soil $CH_4$ production largely occurs in oxygen-depleted soils (Whalen et al., 2005). However, extensive studies of wetland $CH_4$ emissions suggest that inundation is not the sole determinant of spatial and temporal $CH_4$ emission dynamics. $CH_4$ can be transferred directly into the atmosphere via macrophytes, thus circumventing the aerobic soil layer (Whalen et al., 2005). Water-body depth (Mitsch et al., 2010), type (Devol et al., 1990) together with aquatic macrophyte density (Laanbroek 2010) can affect the proportion of wetland $CH_4$ transferred to the atmosphere.

Carbon (C) availability is also a determinant of wetland $CH_4$ emissions. Methanogen-available C turnover rates (Miyajima et al., 1997), composition (Wania et al., 2010), temporal dynamics (Bloom et al., 2012) and C stocks together drive spatial and temporal variability of carbon limitation on $CH_4$ production in wetlands. C cycle state variables, including the spatial variability of total biomass (Saatchi et al., 2011; Baccini et al., 2012) and soil carbon (Hiederer and Köchy, 2011) vary at <1000km scales. Methanogen-available C sources – such as gross primary production (GPP) and leaf litter –vary substantially at monthly timescales in the wet tropics (Beer et al., 2010; Chave et al., 2010; Caldararu et al., 2012). In the next section, we establish the $CH_4$ flux resolution and precision requirements based on the variability of potential tropical wetland $CH_4$ emissions process controls (namely carbon uptake, live biomass and dead organic matter stocks, inundation and precipitation).

## 2.2 Wetland $CH_4$ flux requirements

Here we define a set of wetland $CH_4$ flux precision and resolution requirements suitable for the formulation and testing of wetland $CH_4$ emissions process control hypotheses. Measurement and model-based analyses of Amazon wetland $CH_4$ emissions provide a range of contradictory estimates on spatial patterns and seasonality (Devol et al., 1990; Riley et al., 2011; Bloom et al., 2012; Melton et al., 2013; Basso et al., 2016) suggesting that the basin-wide process controls on wetland

CH$_4$ emissions remain virtually unknown. Here, our aim is to provide a first order, model-independent characterization of wetland CH$_4$ flux resolution and precision requirements based on the basin-wide variations in carbon and hydrological processes. Our resolution requirement is based on the correlation lengths of hypothesized wetland CH$_4$ emission process controls. At the required resolution, our precision requirement is that wetland CH$_4$ emissions scenarios – derived from a
range of hypothesized carbon and hydrological process controls – are (a) statistically inter-distinguishable and (b) distinguishable from a spatio-temporally uniform wetland CH$_4$ flux (i.e. a null hypothesis).

Given our process-level understanding of wetland CH$_4$ emissions, we propose four carbon and three hydrological proxies as the dominant drivers of wetland CH$_4$ emission variability (C1-C4 and H1-H3 respectively). We use carbon stocks and fluxes
as proxies for variation in C availability for wetland CH$_4$ production. We characterize the spatial variability of carbon uptake based on the Jung et al., (2009) eddy-covariance based monthly $0.5° \times 0.5°$ GPP product (C1), and monthly $0.5° \times 0.5°$ solar-induced fluorescence retrieved from the Global Ozone Monitoring Experiment measurements (Joiner et al., 2013; C2). We use the Saatchi et al., (2011) biomass map (C3) and the Hiederer and Köchy, (2011) live biomass and dead organic matter carbon stocks (C4). We define the spatial variability of hydrological controls over methane flux based on two
inundation fraction datasets (Prigent et al., 2012; Schroeder et al., 2015; H1 and H2) and the NASA Tropical Rainfall Measuring Mission (TRMM; Huffman et al., 2007) precipitation retrievals (H3).

*CH$_4$ flux resolution*

Our resolution requirement is based on a first-order assessment of the process variable correlation length scales: we anticipate that retrieving wetland CH$_4$ fluxes at much finer scales may be redundant, while retrieving fluxes at much coarser scales may hinder the potential to investigate biogeochemical process controls on wetland CH$_4$ emission variability. We use an auto-correlative approach to identify the variability length-scales of potential CH$_4$ emissions process controls (see Appendix A). The spatial auto-correlation coefficients (Moran's I) of the seven limiting process variables indicate coherent
spatial structures spanning up to ~ 333km – 666km across the Amazon river basin (Figure 3): process variables exhibit high

auto-correlation at a 1° × 1° resolution ($L \sim 111$km), and no significant spatial correlation at 6° × 6° ($L \sim 666$km). Based on our correlative analysis, we expect that wetland $CH_4$ flux estimates at 3° × 3° ($L \sim 333$km) will likely be critical for a first-order distinction between the roles of carbon and water processes on Amazon wetland $CH_4$ emissions: we propose a ~333km $CH_4$ flux resolution as the spatial resolution required to determine the role of process control variability on wetland $CH_4$

emissions. For all time-varying datasets (C1, C4, H1, H2 and H3), we conducted a lagged Pearson's correlation analysis: the time varying datasets indicate varying levels of statistically significant 1-month auto-correlations across the study region (percent of area exhibiting significant autocorrelations: C1 = 98%; C4 = 6%; H1 = 47%; H2 = 51%; H3 = 64%), while virtually 0% of the study region exhibits significant 2-months temporal auto-correlations. For this study, we opt for a monthly temporal resolution requirement: however, we note that higher-temporal resolution datasets (given their availability)

can potentially provide an improved assessment of the temporal correlation scales of carbon and hydrological process controls.

*$CH_4$ flux precision*

We next derive the $CH_4$ flux precision required to distinguish between hypothesized wetland $CH_4$ process controls at a ~333km monthly resolution. We derive the precision requirements assuming one continuous year of $CH_4$ flux retrievals. We formulate (a) spatial $CH_4$ emission hypotheses, where wetland $CH_4$ emissions linearly co-vary with the hypothesized processes at ~333km scales, and (b) temporal $CH_4$ emission hypotheses, where wetland $CH_4$ emissions linearly co-vary with the hypothesized processes on monthly timescales scales. Our motivation for evaluating both spatial and temporal

hypotheses is that we do not necessarily expect the spatial and temporal process controls on wetland $CH_4$ emissions to be the same. For example, Amazon wetland $CH_4$ emissions could be spatially limited by carbon uptake (GPP) and temporally driven by inundation. Each wetland hypothesis is scaled to an annual mean flux of 12 mg m$^{-2}$ day$^{-1}$, which corresponds to the Melack et al., (2004) annual Amazon-wide wetland $CH_4$ emission estimate (29.3 Tg $CH_4$ yr$^{-1}$ across 668 Mha). The explicit formulation of spatial and temporal wetland $CH_4$ emission hypotheses is described in Appendix B.

For a range of retrieved $CH_4$ flux precisions across the Amazon basin (spanning $1 - 100$ mg m$^{-2}$ day$^{-1}$), we test whether each spatial and temporal wetland $CH_4$ emission hypothesis is statistically distinct from alternative hypotheses and a "no variability" hypothesis (i.e. a null hypothesis); the derivation of the statistical confidence in distinguishing between hypotheses is described in Appendix B. The distinction confidence (%) for spatial and temporal hypotheses is shown in

Figure 4: at a monthly ~333km resolution, both spatial and temporal wetland $CH_4$ emission hypotheses are inter-distinguishable with >95% confidence at a $\leq 10$ mg m$^{-2}$ day$^{-1}$ $CH_4$ flux precision.

*$CH_4$ requirements*

Given the spatial and temporal variability of potential hydrological and carbon controls, we define the following requirements for wetland $CH_4$ flux retrievals:

- $CH_4$ flux spatial resolution = ~333km

- $CH_4$ flux temporal resolution: monthly

- $CH_4$ flux precision: = 10 mg $CH_4$ m$^{-2}$ day$^{-1}$

Our resolution and precision requirements provide a first-order assessment of the wetland $CH_4$ emission biogeochemical process control variability. We anticipate that satellite-based $CH_4$ flux estimates meeting the above-stated requirements will provide robust characterization of spatial variation in Amazon wetland $CH_4$ emissions on the scale of variation in the major

carbon and water controls, allowing forcing (hydrology and carbon) and response ($CH_4$ flux) to be related directly. Therefore, by retrieving $CH_4$ fluxes at the required resolution and precision, carbon and hydrological process hypotheses on the dominant drivers of Amazon wetland $CH_4$ emissions can be adequately investigated. However, depending on the nature of the scientific investigation, we recognize that the trade-off space between spatial resolution, temporal resolution, precision and study duration can be further explored to derive an optimal combination of $CH_4$ flux requirements.

Throughout the next subsections, we characterize the required satellite column $CH_4$ measurements needed to resolve $CH_4$ flux with the above-stated requirements. To quantify the sensitivity of our results to the above-mentioned requirements, we repeat our analysis for a range of $CH_4$ flux spatial resolution requirements ($L$ = 150km – 990km) and we derive the corresponding $CH_4$ flux precision requirements.

**2.3 $CH_4$ observation requirements**

We define the atmospheric $CH_4$ observation requirements by retrieving $CH_4$ fluxes from a range of low-earth orbit (LEO), and geo-stationary orbit (GEO) OS simulated $CH_4$ retrieved concentrations, or "observations". Our approach is three-fold:

(a) we simulate LEO and GEO $CH_4$ observations for March 2007; (b) we derive the precision of $CH_4$ measurement averaged at an $L \times L$ resolution (henceforth the "cumulative $CH_4$ measurement precision"), and (c) we employ an idealized inversion to simulate $CH_4$ flux retrieval uncertainty for March 2007 based on the cumulative $CH_4$ measurement precision. We note that wetland emissions are the largest and most uncertain source of $CH_4$ within the Amazon river basin (Wilson et al., 2016; Melton et al., 2013). We henceforth assume that the non-wetland $CH_4$ contribution (namely fires and anthropogenic $CH_4$

sources) can be relatively well characterized using ancillary datasets and global inventories (Bloom et al., 2015; Turner et al., 2015 and references therein).

*LEO and GEO $CH_4$ observations*

The advantage of LEO systems is a near-global coverage; for the TROPOMI mission $CH_4$ orbit and measurement parameters, this equates to a 1-day maximum re-visit period globally. While a GEO system can only view a fixed area on the globe, revisit periods can be far shorter. To relate $CH_4$ observation requirements to current technological capabilities, we explore six OS configurations based on LEO and GEO OS parameters used to simulate the up-coming GEOCAPE and TROPOMI missions' observations in North America by Wecht et al., (2014) (Table 1). We note that, for regional $CH_4$

emission estimates, the GEO OS configurations are expected outperform LEO due to a larger data volume: the fixed viewing

area permits multiple re-visits per day (Wecht et al., 2014), and the smaller GEO footprint size typically leads to lower cloud-contamination (Crisp et al., 2004). Our aim here is not to compare $CH_4$ emission estimates from LEO and GEO $CH_4$ retrievals. Rather, our aim is to determine whether $CH_4$ emission estimates from a range of LEO and GEO OS configurations are able meet the wetland process requirements outlined in section 2.1.

Cloud cover is a major limiting factor in Amazon basin trace-gas retrievals. Mean March 2007 cloud cover is 89% – ranging from 38% to 98% at a 1° × 1° resolution – throughout the Amazon river basin (based on MODIS cloud-cover data, Figure B1). We quantify the data-rejection due to cloud cover based on 1km March 2007 MODIS cloud cover data. Based on four MODIS cloud cover flags, we categorize 1km × 1km cloud-cover observations into "cloud-contaminated" and "cloud free"

observations (see Appendix C). Any cloud-contaminated 3km×3km (GEO) or 7km×7km (LEO) $CH_4$ measurement footprints are rejected, i.e. all accepted footprints are 100% "cloud-free".

To assess the relative importance of $CH_4$ measurement density in high cloud-cover areas, we test two additional geo-stationary configurations: "GEO-Z1" carries out two visits per day and 6 visits per day in the top 50% cloudiest areas;

"GEO-Z2" carries out two visits per day and 10 visits per day in the top 25% cloudiest areas (we note that these two OS would require targeting capabilities to optimize the sampling strategy over the cloudiest area of the basin). We further explore OS space by testing LEO with a $\sqrt{2}$ precision enhancement ("LEO+") and GEO with 8 visits per day instead of 4 ("GEO×2").

*Cumulative $CH_4$ measurement precision*

For each OS $\omega$ ("GEO","LEO", etc.), $\mathbf{O}^{\{L,\omega\}}$ is the cumulative $CH_4$ measurement precision at a $L \times L$ resolution. $\mathbf{O}^{\{L,\omega\}}$ is an $N \times 1$ array, where $N$ is the number of Amazon river basin grid-cells at resolution $L \times L$. We derive the cumulative atmospheric $CH_4$ precision within each $L \times L$ grid-cell $i$, $O_i^{\{L,\omega\}}$ as follows:

$$O_i^{\{L,\omega\}} = \frac{\sigma_\omega}{\sqrt{a\,\phi_i^{\{\omega\}}\,n^{\{\omega\}}\,L^2}} \qquad (1)$$

where $\sigma_\omega$ is the single observation precision (table 1), $\phi_i^{\{\omega\}}$ is the fraction of cloud-free observations at location $i$, $n^{\{\omega\}}$ is the number of observations per km$^2$ per month for OS $\omega$ (based on Table 1 values), and $a$ the fraction of accepted cloud-free CH$_4$ column retrievals (set to $a = 0.5$); The derivation of $\phi_i^{\{\omega\}}$ is based on MODIS 1-km cloud cover data over the Amazon river basin in March 2007 (Appendix C). The square of the denominator in (1) corresponds to the number of atmospheric column CH$_4$ measurements per $L \times L$ grid-cell. For all OS, $n^{\{\omega\}}$ is calculated assuming continuous basin-wide coverage at the single-sounding footprint resolution (see Table 1). We highlight that our formulation of cumulative CH$_4$ precision in equation 1 implies retrieved CH$_4$ errors are spatially and temporally uncorrelated.

*OS retrieved CH$_4$ flux precision*

We calculate the monthly retrieved CH$_4$ flux precision for OS $\omega$ at an $L \times L$ resolution – $\mathbf{F}^{\{L,\omega\}}$ –based on $\mathbf{O}^{\{L,\omega\}}$ (equation 1). $\mathbf{F}^{\{L,\omega\}}$ is a $N \times 1$ array, where $N$ is the number of Amazon basin grid-cells at resolution $L \times L$. To calculate $\mathbf{F}^{\{L,\omega\}}$ we simulate an ensemble of 1000 retrieved CH$_4$ concentrations vectors ($\mathbf{c}_{*,n}^{\{L,\omega\}}$ for $n = 1 - 1000$) over the Amazon river basin, where:

$$\mathbf{c}_{*,n}^{\{L,\omega\}} = \mathbf{c}^{\{L,0\}} + \mathbf{N}(0,1) \circ \mathbf{O}^{\{L,\omega\}}; \qquad (2)$$

$\mathbf{c}^{\{L,0\}}$ is a $N \times 1$ array of $L \times L$ gridded unperturbed CH$_4$ concentrations, $\mathbf{N}(0,1)$ is an $N \times 1$ array of normally distributed random numbers with mean zero and variance one ("$\circ$" denotes element-wise multiplication). We relate the concentrations $\mathbf{c}^{\{L,*\}}$ to the underlying CH$_4$ fluxes $\mathbf{f}^{\{L,*\}}$ as follows:

$$\mathbf{c}^{\{L,*\}} = \mathbf{A}^{\{L\}}\mathbf{f}^{\{L,*\}}, \qquad (3)$$

where $\mathbf{A}^{\{L\}}$ is the atmospheric transport operator (the $N \times N$ matrix transforming fluxes to concentrations over the Amazon river basin domain) and $\mathbf{f}^{\{L,*\}}$ is an $N \times 1$ array of surface $CH_4$ fluxes. For the sake of brevity, we present a summary of $\mathbf{A}^{\{L\}}$ here, and the complete derivation of $\mathbf{A}^{\{L\}}$ in Appendix D. We use a Lagrangian Particle Dispersion Model (LPDM: Uliasz,

1994; Lauvaux and Davis, 2014) to derive an "influence function" (or "column footprint") relating satellite-retrieved atmospheric $CH_4$ concentrations to surface fluxes (the inverse solution of the transport from the surface to higher altitudes) at the center of the study area. We simulate 30km $\times$ 30km $CH_4$ transport – $\mathbf{A}^{\{30km\}}$ – by spatially translating the LPDM influence function throughout the domain. To assess the robustness of the LPDM approach, we also simulated $CH_4$ column mixing ratios over the Amazon river basin at 30km using the Weather Research and Forecasting model (WRF v2.5.1,

Skamarock et al., 2008). The WRF model March 2007 Amazon river basin concentrations and the corresponding LPDM approximations are shown in Figure D1. Finally, we used a Monte Carlo approach to statistically construct $\mathbf{A}^{\{L\}}$ based on $\mathbf{A}^{\{30km\}}$. The LPDM, WRF and the Monte Carlo derivation of $\mathbf{A}$ are fully described in Appendix D.

For each $L$, we simulate the flux uncertainty based on the inverse of $\mathbf{A}^{\{L\}}$, $(\mathbf{A}^{\{L\}})^{-1}$ and simulated $CH_4$ concentrations vectors

($\mathbf{c}_{*,n}^{\{L,\omega\}}$, equation 2). For the sake of simplicity, we set all unperturbed concentrations – $\mathbf{c}^{\{L,0\}}$ in equation 2 – to be equal to zero, since these do not influence our subsequent derivation of $\mathbf{F}^{\{L,\omega\}}$. The $n^{th}$ retrieved flux estimate – $\mathbf{f}_{*,n}^{\{L,\omega\}}$ – is calculated as:

$$\mathbf{f}_{*,n}^{\{L,\omega\}} = (\mathbf{A}^{\{L\}})^{-1} \, \mathbf{c}_{*,n}^{\{L,\omega\}}. \tag{4}$$

Finally, we calculate the flux precision $\mathbf{F}^{\{L,\omega\}}$ at grid-cell $i$ as follows:

$$F_i^{\{L,\omega\}} = StDev\left(\mathbf{f}_{i,*}^{\{L,\omega\}}\right). \tag{5}$$

Despite the implementation of CH$_4$ bias correction methods based on satellite CH$_4$ retrieval comparison against ground measurements of total column CH$_4$ (Parker et al., 2011), spatial structures in residual CH$_4$ biases are a key limiting factor in

top-down CH$_4$ flux accuracy. Here we quantify the role residual CH$_4$ biases for each OS configurations. We simulate a retrieved pseudo-random CH$_4$ bias structure with a spatial correlation of $s = 100$km and no temporal correlation, which is consistent with the likely first order predictors of retrieved CH$_4$ residual biases (Worden et al., 2016). Here we simulate a range of pseudo-random bias distributions with standard deviations spanning $b = 0.5 - 50$ppb. For each $b$, we calculate the bias-influenced flux uncertainty $\mathbf{F}^{\{L,\omega,b\}}$ based on equations 4 and 5: to incorporate spatially correlated biases, we adapt eq. 2

to derive the mean concentration uncertainty $\mathbf{c}'^{\{L,\omega,b\}}_{*,n}$ as

$$\mathbf{c}'^{\{L,\omega,b\}}_{*,n} = \mathbf{N}(0,1) \cdot \mathbf{O}^{\{L,\omega\}} + \mathbf{N}(0,1) \cdot b \cdot \frac{s}{L\sqrt{v}} \qquad (6)$$

where $b$ represents the standard deviation of the pseudo-random CH$_4$ bias, $v$ represents the number of visits per month; for

bias errors correlated across spatial scales $s$, the scale factor $\frac{s}{L\sqrt{v}}$ accounts for the pseudo-random behaviour of bias errors $b$ at a monthly $L \times L$ resolution. We assess the role CH$_4$ biases on $\mathbf{F}^{\{L,\omega,b\}}$ for the LEO and GEO OS configurations at $L = $ ~333km.

## 3. Results and Discussion

Cumulative CH$_4$ precision for mean monthly atmospheric column CH$_4$ measurements is $0.10 - 0.98$ ppb for the LEO OS (Figure 5, left) and $0.02 - 0.20$ ppb for the GEO OS (Figure 5, right). The lowest CH$_4$ concentration precision occurs in the East and central Amazon river basin. A crucial advantage of the smaller GEO OS footprint is the 88–148% higher probability of cloud-free observations in the cloudiest regions of the Amazon river basin (Figure B1); the probability of

acquiring cloud-free observations in cloud-prone areas is further enhanced by the GEO OS ability to conduct multiple visits per day (see eq. 1).

For $L$ = ~333km, median monthly retrieved $CH_4$ flux precision for the LEO OS (i.e. the median of $\boldsymbol{F}^{\{L,\omega\}}$) is 17.0 mg $CH_4$ $m^{-2}$ $day^{-1}$ (Figure 6); increasing the single sounding retrieval precision by $\sqrt{2}$ (from 0.6ppb to 0.42ppb) for LEO observations (LEO+) reduces the retrieved flux uncertainty to 11.9 mg $CH_4$ $m^{-2}$ $day^{-1}$. This uncertainty reduction is equivalent to a second LEO visit per day (see table 1): the factor 3-to-10 lower uncertainties for cumulative GEO $CH_4$ concentrations (Figure 5) lead to a 2.7 mg $CH_4$ $m^{-2}$ $day^{-1}$ median uncertainty in the retrieved flux (Figure 6). Doubling the number of GEO visits per day (GEOx2 OS) reduces the retrieved flux uncertainty to 1.9 mg $CH_4$ $m^{-2}$ $day^{-1}$. GEO-Z1 and GEO-Z2 uncertainties (2.4 and 2.0 mg $CH_4$ $m^{-2}$ $day^{-1}$) are both lower than GEO. These results indicate that – despite a lower number of accepted observations – a higher observation density in the high cloud-cover areas of the Amazon basin (and lower observation density elsewhere) can be used to reduce the retrieved $CH_4$ flux uncertainty without increasing the number of observations per day. Based on the LEO OS, we anticipate that missions similar to the ESA TROPOMI observation configuration (Veefkind et al., 2012; Wecht et al., 2014) will lead to lower-than-required information content for Amazon wetlands and are unlikely to provide sufficient observational constraints to resolve the dominant $CH_4$ flux processes.

Our bias $CH_4$ analysis (Figure 7) indicates that GEO retrieved $CH_4$ flux precisions at $L$ = ~333km are relatively unaffected by residual $CH_4$ biases <1ppb, while LEO retrieved $CH_4$ flux precisions are relatively unaffected by residual $CH_4$ biases <5ppb. We find that the advantage of GEO $CH_4$ flux precision over LEO diminishes from almost one order of magnitude at residual $CH_4$ biases <1ppb, to roughly a factor of 2 for residual biases >20ppb. Here we assume a residual $CH_4$ bias correlation scale of 100km (section 2.3); based on eq. 6, we expect a larger impact of residual $CH_4$ biases on OS retrieved $CH_4$ flux precision for residual $CH_4$ bias correlation lengths >100km or for temporally correlated $CH_4$ biases. Overall, the relative advantage of GEO over LEO OSs is contingent on both the cumulative $CH_4$ precision (Figure 5) as well as the anticipated spatiotemporal structure of residual $CH_4$ bias.

Estimates of fluxes at $L = 150 – 990$km show that median GEO retrieved $CH_4$ flux uncertainty is consistently a factor of ~5 lower than the median LEO retrieved $CH_4$ flux uncertainty (Figure 8); for a 10ppb residual pseudo-random bias, the median GEO retrieved flux uncertainty is consistently a factor of ~3 lower than LEO-retrieved flux uncertainty. GEO-derived $CH_4$ fluxes meet the both the precision and resolution requirements for $L = $ ~180 – 333km; for a 10ppb residual bias, GEO-derived $CH_4$ fluxes meet both requirements at $L = $ ~280 – 333km. At the expense of the resolution requirement, both GEO simulations meet the precision requirements for all $L \geq$ ~333 km. Unbiased median LEO-derived $CH_4$ fluxes meet the precision requirements at $L>500$km; LEO-derived $CH_4$ fluxes with a 10ppb pseudo-random bias meet the precision requirement at $L>800$km and partially meet the precision requirement for 550km$>L>800$km.

In our analysis we have assumed (i) no systematic biases in our atmospheric inversion simulation, and (ii) perfectly known boundary conditions. Significant systematic atmospheric $CH_4$ retrieval and transport model biases can undermine the enhanced accuracy of geostationary OSs. For example, we find that our LPDM-derived transport operator yields a conservative estimate of the monthly mean $CH_4$ gradient across the domain relative to the WRF model simulation (Appendix D; Figure D1). We assess the sensitivity of our results to a factor of 1.5 increase in the LPDM-derived transport operator ($\mathbf{A}^{\{L\}}$); OS $CH_4$ flux precision results exhibit an inversely proportional response, corresponding to a ~33% uncertainty reduction (median GEO flux precision of 1.8 mg $CH_4$ m$^{-2}$ day$^{-1}$ and a LEO precision of 11.3 mg $CH_4$ m$^{-2}$ day$^{-1}$). GEO missions are likely to provide a higher volume of observations at the boundaries of the observation domain, relative to LEO OS: therefore, boundary conditions are likely to reinforce the potential of GEO OS compared to LEO. We recognize that further efforts are required to fully assess the role of seasonal transport variability, transport errors, boundary condition assumptions and atmospheric $CH_4$ bias structures on the accuracy of GEO and LEO $CH_4$ flux retrievals.

We note that a limiting factor in our analysis is the lack of data constraints on diurnal cloud-cover variability (since the MODIS cloud cover dataset does not provide diurnal constraints). The March 2007 ERA-interim monthly mean 3h cloud cover dataset indicates a 7 – 80% (median 29%) coefficient of variation of cloud-free fraction diurnal variability throughout the Amazon basin. Given the non-linear sensitivity of data yield to synoptic cloud cover (Figure B1), the cloud-free fraction

coefficient of variation may amount to an important component in assessing and optimizing the performance of LEO and GEO OSs over the Amazon basin, as well as other high cloud-cover regions across the globe.

Our $CH_4$ flux resolution requirement (monthly $L = \sim333km$ $CH_4$ flux retrievals) is derived based on an assessment of carbon
and hydrological auto-correlation scales across the Amazon river basin. Although our sensitivity analysis (Figure 8) shows that GEO can potentially distinguish between the hypothesized $CH_4$ emission scenarios at $L > \sim333km$, we anticipate that additional biogeochemical investigations – such as the second-order interactions between carbon and hydrological drivers on wetland $CH_4$ emissions – would likely be increasingly challenging at coarser resolutions. We recognize that our resolution requirement and our quantification of correlation scales is specific to our study region: for example, quantification of
greenhouse gas measurement requirements for finer-scale studies would yield a unique set of requirements, and supporting analyses may require higher-resolution datasets. Our approach provides the means to examine trade-offs between spatial and temporal resolutions. For example, further analyses can be conducted to establish the space-time trade-offs to optimize biogeochemical investigations and process uncertainty reduction. We also note that GEO OSs provide unprecedented volume of observations: the enhanced sampling approach can potentially be used at shorter timescales to optimally resolve
source and transport patterns. This approach could be particularly useful in instances wetland $CH_4$ emissions are densely focused in space or time. Finally, we highlight the potential for combining multiple OSs (e.g. LEO and GEO systems) to optimally constrain $CH_4$ fluxes and biogeochemical process controls; the potential of OS synergies undoubtedly requires further investigation.

In contrast to our approach, $CH_4$ flux uncertainty requirements can alternatively be derived by quantifying process-based wetland $CH_4$ emission model uncertainty (Melton et al., 2013), or by characterizing the $CH_4$ flux uncertainty stemming from wetland $CH_4$ model parametric uncertainty (Bloom et al., 2012). An advantage of model-based requirements is the ability to assess $CH_4$ flux uncertainties associated with the complex interactions between wetland $CH_4$ processes (e.g. Riley et al., 2011). Prior information on the magnitude and variability of fluxes can also be introduced (e.g. in a Bayesian atmospheric
transport and chemistry inversion framework) to re-assess posterior uncertainty estimates.

However, as outlined in section 2.1, large unknowns preside over the processes governing the spatial and temporal variability of wetland $CH_4$ fluxes. Moreover, wetland $CH_4$ models often exhibit structural similarities (Melton et al., 2013); for example, wetland $CH_4$ emission models (Melton et al., 2013) suggest major $CH_4$ emissions along the main stem of the Amazon river (Figure 1). Since model spatiotemporal $CH_4$ flux variations – and their associated processes – have not been adequately assessed due to insufficient in-situ measurements (particularly in the tropics), the introduction of prior spatial and temporal correlations in wetland $CH_4$ flux estimates would hinder the potential to independently investigate biogeochemical process controls on wetland $CH_4$ emissions. To our knowledge, our analysis provides a first quantification of the OS requirements for confronting prior knowledge on $CH_4$ fluxes at a process-relevant resolution.

## 4. Concluding remarks

Quantitative knowledge of biogeochemical processes controlling biosphere-atmosphere greenhouse gas fluxes remains highly uncertain. Optimally designed satellite greenhouse gas observing systems can potentially resolve the processes controlling critical boreal and tropical greenhouse gas fluxes. In this study, we have characterized a satellite OS able to resolve the principal process controls on Amazon river basin wetland $CH_4$ emissions. Conventional low-earth orbit satellite missions will likely be unable to resolve Amazon wetland $CH_4$ emissions at a process-relevant scale and precision. Observation density in time and space, and its reduction by cloud cover are the major limiting factors. Increasing the number of daily $CH_4$ measurements in cloudy regions at the expense of other measurements can further reduce the retrieved $CH_4$ flux precision from geostationary satellite $CH_4$ measurements. OSSEs based on reducing process uncertainty can inform observation requirements for future greenhouse gas satellite missions in a far more targeted way than simply quantifying overall flux uncertainty reduction for a given OS.

**Appendix A: Correlation lengths**

All datasets described in section 2.2 were aggregated to a common $0.5° \times 0.5°$ resolution. For each process control dataset, we derive the Moran's I spatial auto-correlation coefficient $(r_{MI})$ at an $L \times L$ resolution, where $L = 0.5°, 1°, 1.5°, \dots, 10°$. For every $L$ we aggregated the dataset to $L \times L$ resolution. To determine whether the derived $r_{MI}$ are significant relative to the null hypothesis, we repeat the Moran's I derivation 2000 times for normally distributed random numbers (in the place of the $L \times L$ gridded dataset), which together statistically represent the Moran's I distribution $(\mathbf{R}_{MI})$ for statistically insignificant spatial correlation. When $r_{MI} > \text{median}(\mathbf{R}_{MI})$, the $r_{MI}$ p-value is twice the fraction of instances where $\mathbf{R}_{MI} > r_{MI}$; when $r_{MI} < \text{median}(\mathbf{R}_{MI})$, the $r_{MI}$ p-value is twice the fraction of instances where $\mathbf{R}_{MI} < r_{MI}$. A p-value $\geq 0.05$ indicates that the null hypothesis cannot be rejected with a 95% confidence.

**Appendix B: Spatial and temporal wetland CH$_4$ emission hypotheses**

*Detectability of wetland CH$_4$ hypotheses*

Based on the four carbon and three hydrological proxies (see section 2.2), we formulate spatial and temporal wetland CH$_4$ emission hypotheses (henceforth **S** and **T** respectively) – at a monthly ~333km × 333km resolution – and determine our ability to statistically distinguish between these at a range of retrieved CH$_4$ flux precisions ($p = 1 - 100$ mg m$^{-2}$ day$^{-1}$). For all **S**, we prescribe temporally constant CH$_4$ emissions and for **T** we annually normalize mean annual emissions to 12 mg m$^{-2}$ day$^{-1}$ within each ~333km × 333km area; For both **S** and **T** we also include a "no variability" scenario, where all emissions in space and time are 12 mg m$^{-2}$ day$^{-1}$. We note that by minimizing the variability of each hypothesis to a single temporal or spatial variable, we effectively assume a "worst-case" scenario for the detectability **S** and **T** hypotheses relative to the null hypothesis.

For hypothesized process control $h$ we derive the temporal wetland CH$_4$ emission hypothesis $\mathbf{T}_{*,*,h}$, as:

$$T_{x,t,h} = s_x \, P_{x,t,h} \tag{B1}$$

where $P_{x,t,h}$ represents the hypothesized process control $h$ at location $x$ and time $t$, and $s_x$ is a scaling factor such that $\overline{T_{x,*,h}}$ = 12 mg m$^{-2}$ day$^{-1}$. For the temporal hypotheses we omit the soil carbon and carbon stock proxies, as these datasets are not temporally resolved. Each spatial hypothesis $\mathbf{S}_{*,*,h}$ is defined as

$$S_{x,t,h} = s\,\overline{P_{x,*,h}} \tag{B2}$$

where $s$ is a scaling factor such that $\overline{S_{*,t,h}}$ = 12 mg m$^{-2}$ day$^{-1}$. For each hypothesis $h$ and each precision $p$ we simulate retrieved CH$_4$ fluxes $F_{x,t,h,p}$ as

$$F_{x,t,h,p} = H_{x,t,h} + N(0,1)\cdot p \tag{B3}$$

where $H_{x,t,h}$ is the spatial or temporal hypothesis CH$_4$ flux ($T_{x,t,h}$ or $S_{x,t,h}$) and N(0,1) is a normally distributed number with mean 0 and variance of 1. For each $h$, we compare $\mathbf{F}_{*,*,h,p}$ against all hypothesized process controls $h'$ as follows:

$$J_{h,h',p} = \sum_{x,t}(F_{x,t,h,p} - H_{x,t,h'})^2 \tag{B4}$$

We repeat the derivation of $\mathbf{J}$ 500 times, and we define the detectability confidence $C_{h,p}$ as the percentage of times where $J_{h,h,p} = \min(\mathbf{J}_{h,*,p})$; the min() function denotes the minimum of all $\mathbf{J}_{h,*,p}$ elements. In summary, $C_{h,p}$ is the probability of distinguishing a hypothesized wetland CH$_4$ process control $h$ from alternative wetland CH$_4$ process controls when wetland CH$_4$ fluxes are retrieved with precision $p$. $C_{h,p}$ values for spatial and temporal wetland CH$_4$ hypotheses are summarized in Figure 4. We henceforth define a wetland CH$_4$ hypothesis as "distinguishable" from alternative hypotheses at precision $p$ when $C_{h,p} > 95\%$.

**Appendix C: MODIS cloud cover**

The MODIS cloud cover analysis was performed based on the MOD06_L2 1km cloud mask product (downloaded from modis.gsfc.nasa.gov). We consider "probably cloudy" and "cloudy" 1km × 1km pixel flags as cloud-covered areas (CC = 1), and the remaining pixel flag categories ("probably clear" and "clear") as cloud-free areas (CC=0): here we assume that the statistical patterns of cloud-cover across the Amazon domain remain well characterized when assigning "probably clear" and "probably cloudy" pixels to the "cloud-free" and "cloud-covered" categories. We aggregate the 1km data to $N$km × $N$km ($N$ is the OS footprint resolution; GEO N = 3km; LEO N = 7km; see Table 1) to calculate the number of cloud-free $N$km × $N$km areas within each MODIS cloud cover scene. The monthly fraction of cloud-free observations $\phi_i^{\{\omega\}}$ (see equation 1) is calculated by deriving the ratio of cloud-free to total $N$km × $N$km areas within each $L \times L$ area. A regional summary of the observation yields (% of cloud-free $N$km × $N$km areas) for a range of footprint resolutions ($N = 1 - 10$km) is shown in Figure B1.

**Appendix D: Atmospheric transport operator**

For $L$ = 150km - 990km, we derive the $N \times N$ atmospheric transport operator $\mathbf{A}^{\{L\}}$ for $L \times L$ resolution fluxes based on $N$ random $CH_4$ flux vectors ($\mathbf{f}'^{\{L\}}$) and their corresponding concentrations ($\mathbf{c}'^{\{L\}}$): $\mathbf{f}'^{\{L\}}$ and $\mathbf{c}'^{\{L\}}$ are $N \times N$ arrays, where each column of $\mathbf{f}'^{\{L\}}$ is a vector of randomly sampled $CH_4$ fluxes throughout the domain, and each column in $\mathbf{c}'^{\{L\}}$ is a vector of the corresponding $CH_4$ concentrations. $\mathbf{A}^{\{L\}}$ is derived as:

$$\mathbf{A}^{\{L\}} = \left(\mathbf{f}'^{\{L\}}\right)^{-1} \mathbf{c}'^{\{L\}}. \tag{D1}$$

For each $n$, random $CH_4$ fluxes at grid-cell $i$ are derived as $f_{i,n}'^{\{L\}} = R(0,1)$, where $R(0,1)$ is a random number sampled from a normal distribution with mean zero and variance 1. Atmospheric concentrations are firstly simulated at resolution $L_0 = 30$km;

the fluxes $\mathbf{f}'^{\{L\}}_{*,n}$ are downscaled to $L_0 \times L_0$ resolution ($\mathbf{f}'^{\{L_0\}}_{*,n}$). For each 30km × 30km grid-cell $i$, the mean atmospheric CH$_4$ concentration $c^{\{L_0\}}_i$ is calculated as

$$c'^{\{L_0\}}_{i,n} = \mathbf{I}_i \mathbf{f}'^{\{L_0\}}_{*,n} \qquad (D2)$$

where $\mathbf{f}^{\{L_0\}}_{*,n}$ is the N × 1 array of CH$_4$ fluxes, $\mathbf{I}_i$ is the N × 1 influence function array for grid-cell $i$. We derive $\mathbf{I}_i$ using a Lagrangian Particle Dispersion Model (LPDM, Uliasz, 1994). The influence function derivation (i.e. the column sensitivity to the surface fluxes) is described in Lauvaux and Davis (2014). The influence function was computed for an averaged column observation in the model of the simulation domain, for every hour of March 2007. The inverse calculation of surface

10  fluxes requires the use of the adjoint of the transport at the mesoscale (~2000km). Here, we only simulated the fraction of the column influenced by surface fluxes. We assume boundary conditions are well constrained by satellite and surface network measurements: therefore, only the first 6km of the column was described by the particles released backward in the model.

To simulate total column CH$_4$ retrieval influence functions, we incorporate a mean GOSAT CH$_4$ retrieved averaging kernel

15  (Parker et al., 2011) for the Amazon river basin region (Figure A1). To minimize the computational cost of simulating atmospheric transport, we (i) derive the influence function for the center of the domain ($\mathbf{I}_0$, Lat = 4.9°S and Lon = 63.8°W), and (ii) we derive $\mathbf{I}_i$ by spatially translating $\mathbf{I}_0$ to gridcell $i$ latitude and longitude coordinates. Finally, we derive mean $L \times L$ resolution concentrations used in equation C1, ($\mathbf{c}'^{\{L\}}_{*,n}$), based on the spatial aggregation of $L_0 \times L_0$ resolution concentrations $\mathbf{c}'^{\{L_0\}}_{*,n}$.

To assess the viability of our approach, we simulate March 2007 $L_0 \times L_0$ atmospheric concentrations – based on $\mathbf{f}^{\{L_0,0\}}$, where for $i$=1 – N, $f^{\{L_0,0\}}_i$= 12 mg m$^{-2}$ day$^{-1}$ – throughout the Amazon river basin domain using (a) equation D2, and (b) WRF CH$_4$ atmospheric transport model. In the WRF model, $\mathbf{f}^{\{L_0,0\}}$ was coupled to the atmospheric model through the chemistry modules (WRF-Chem) for passive tracers, as described in Lauvaux et al. (2012). The physics configuration of the

model used Mellor-Yamada-Nakanishi-Niino scheme for the Planetary Boundary Layer (Nakanishi and Niino, 2004), the NOAH land surface model (Pan and Mahrt, 1987), the WSM-5 microphysics scheme (Hong et al., 2004), and the Kin-Fritsch cumulus parameterization (Kain, 2004). The meteorological driver data from the Global Forecasting System (FNL) analysis products at 1° × 1° resolution was used at the boundaries of the simulation domain. The simulation domain spans

120x100 $L_0 \times L_0$ grid-points, and 60 vertical levels to describe the atmospheric column up to 50 hPa. The atmospheric column was extracted from the surface to the top of the modeled atmosphere, which represents about 90% of the total air mass. A dilution factor of 0.9 was used to compensate for the partial model column.

The LPDM approach emulates the large-scale WRF $CH_4$ enhancement ($r^2 = 0.85$ see Figure D1); the smoothing effect is due

to the use of a single footprint throughout the entire domain. Mean $CH_4$ concentrations based on our approach (equation D2) and WRF are 15.23ppb and 17.42ppb respectively. The gradient of $CH_4$ between the North-East and South-West sub-regions for our approach (equation D2) and WRF are 13.14ppb and 17.24ppb respectively; the delineation of the North-East and South-West domain is shown in Figure D1.

*Acknowledgments*

*Part of this research was carried out at the Jet Propulsion Laboratory, California Institute of Technology, under a contract with the National Aeronautics and Space Administration. Inundation datasets are available at noaacrest.org/rscg/ and wetlands.jpl.nasa.gov. TRMM data is available at mirador.gsfc.nasa.gov. The gross primary production dataset was obtained from bgc-jena.mpg.de. The soil carbon dataset is available at esdac.jrc.ec.europa.eu. ERA-interim synoptic*

*monthly mean re-analyses were downloaded from apps.ecmwf.int.*

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

**Tables**

**Table 1:** Observation system characteristics[a]

| Observation System | Single sounding footprint size | Single CH$_4$ measurement precision | Visits per day |
|---|---|---|---|
| LEO | 7km × 7km | 0.6% (10.8 ppb) | 1 |
| GEO | 3km × 3km | 0.6% (10.8 ppb) | 4 |
| LEO+ [b] | 7km × 7km | 0.42% (7.6 ppb) | 1 |
| GEO×2 | 3km × 3km | 0.6% (10.8 ppb) | 8 |
| GEO-Z1 | 3km × 3km | 0.6% (10.8 ppb) | 4[c] |
| GEO-Z2 | 3km × 3km | 0.6% (10.8 ppb) | 4[d] |

[a]LEO and GEO observation parameters are broadly consistent with TROPOMI and GEOCAPE simulations by Wecht et al.,(2014); to simplify comparisons, we set GEO and LEO default single CH$_4$ sounding precision to 0.6%.

[b]Single measurement precision is a factor of $\sqrt{2}$ higher than LEO; this is the equivalent to doubling the visits per day for LEO.

[c]2 (6) visits per day in 0 – 50%ile (50 – 100%ile) cloud-cover areas;

[d]2 (10) visits per day in 0 – 75%ile (75 – 100%ile) cloud-cover areas;

**Figures**

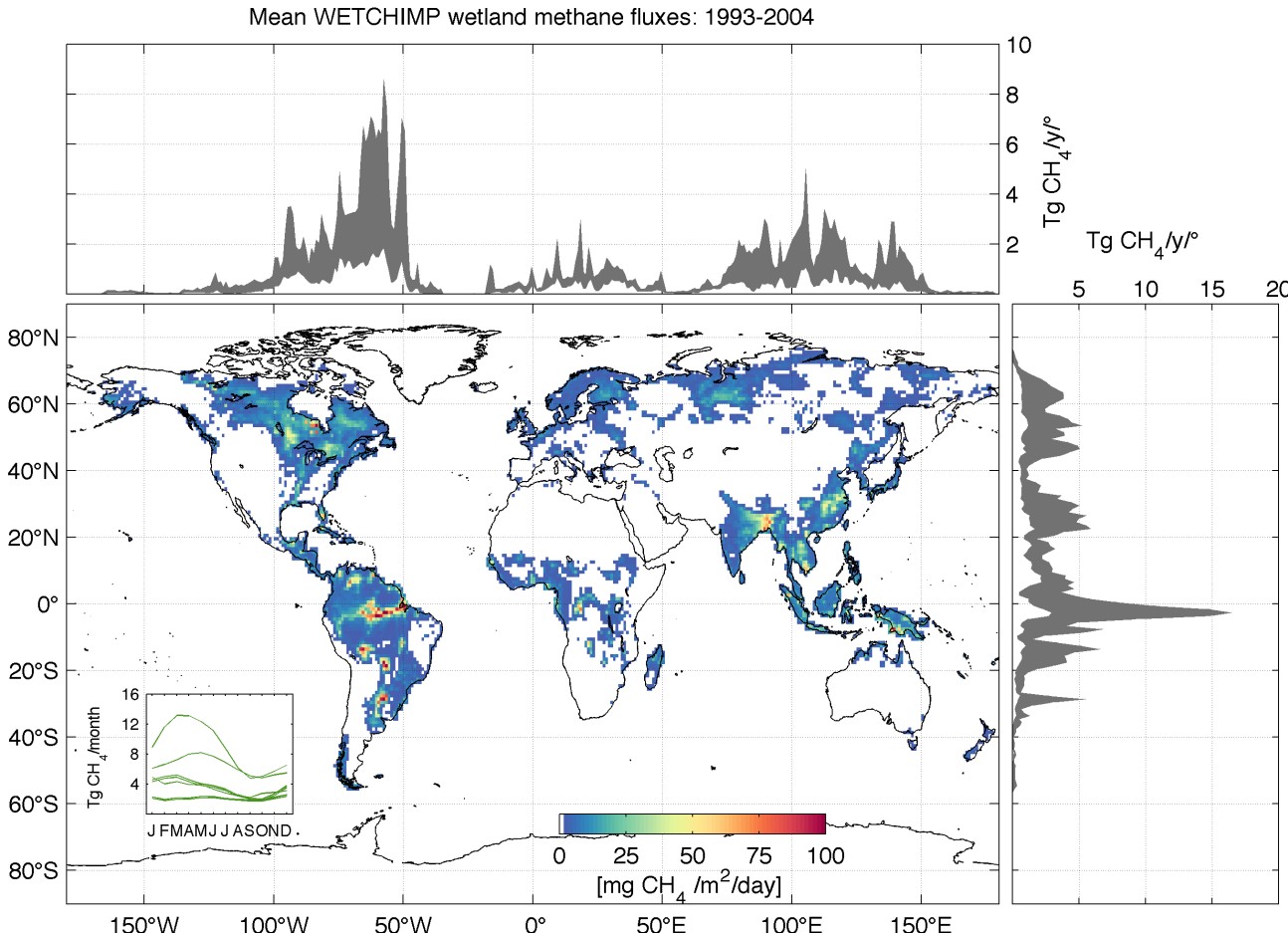

**Figure 1:** Mean annual wetland and rice CH$_4$ emissions (central maps), and associated longitudinal and latitudinal
uncertainty (grey bands), based on the WETCHIMP model inter-comparison project (Melton et al., 2013). **Inset**:
WETCHIMP model total Amazon basin monthly CH$_4$ emissions.

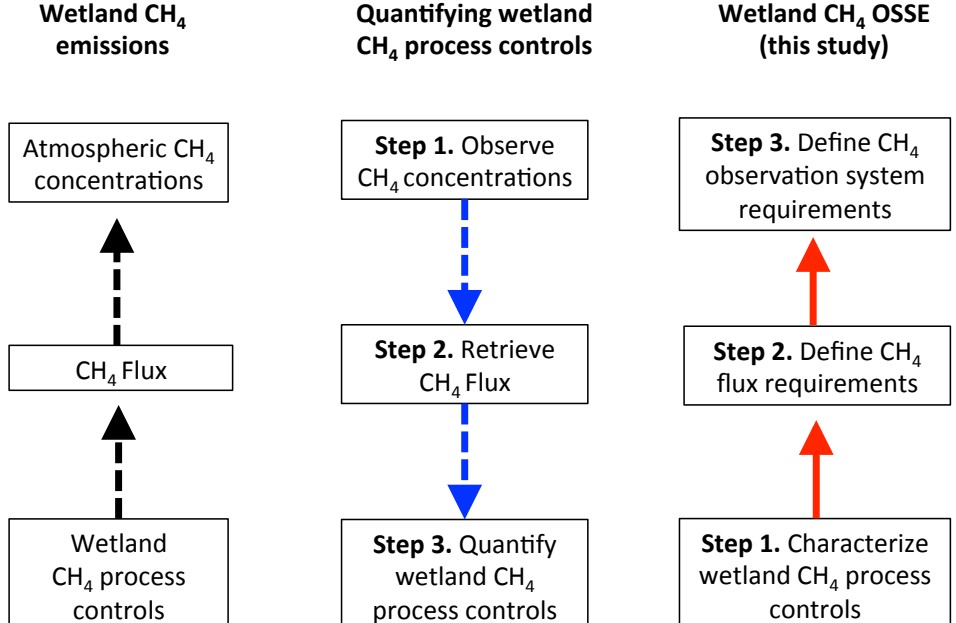

**Figure 2.** Wetland CH$_4$ emissions into the atmosphere are regulated by wetland biogeochemical processes (left column). Continental-scale wetland CH$_4$ process controls can be retrieved by (i) resolving surface CH$_4$ fluxes from retrieved satellite CH$_4$ observations; (ii) resolving process parameters from retrieved CH$_4$ fluxes (middle column). The optimal satellite CH$_4$ observation requirements are a function of the flux resolution and precision required to resolve wetland CH$_4$ process controls (right column): OSSE steps 1-3 are described in sections 2.1-2.3.

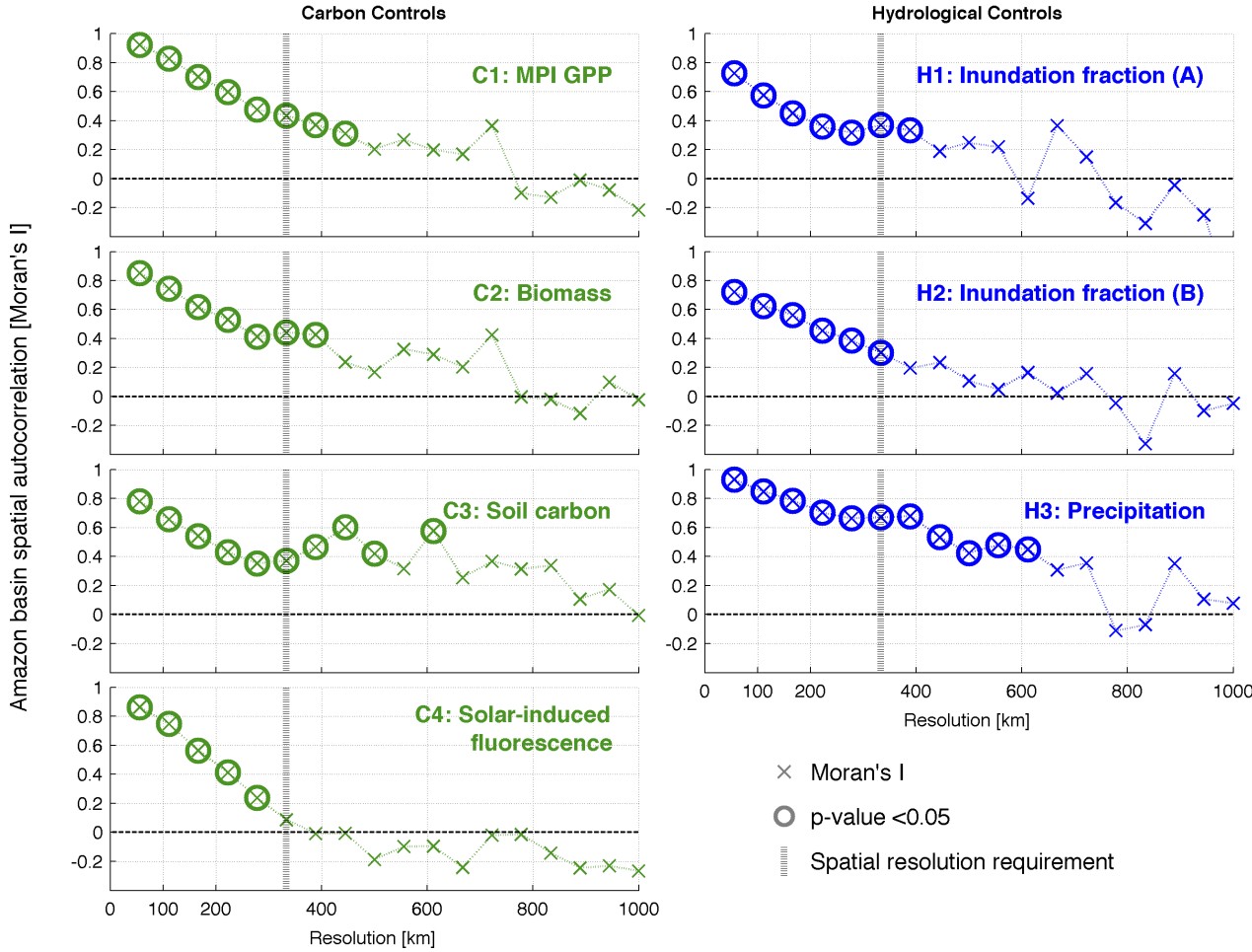

**Figure 3:** Spatial autocorrelation (Moran's I) for potential carbon controls (left column) and hydrological controls (right column) on wetland $CH_4$ emissions. The spatial variability of carbon controls are derived from satellite observations (Biomass, Saatchi et al., 2011; solar induced fluorescence; Joiner et al., 2013), the Harmonized World Soil database (soil carbon, Hiederer & Köchy, 2011) and FLUXNET derived GPP (Jung et al., 2009). The spatial variability estimates for hydrological controls are based on satellite measurements of inundation (A: Prigent et al., 2007; B: Schroeder et al., 2015), and precipitation (the NASA Tropical Rainfall Measuring Mission). Significant Moran's I values (where the Moran's I p-value < 0.05) are highlighted as circles. We set a ~333km spatial resolution requirement for monthly $CH_4$ flux retrievals,

based on the maximum correlation lengths of potential carbon and hydrological controls on wetland CH$_4$ emissions. The details of the Moran's I analysis are fully described in Appendix A.

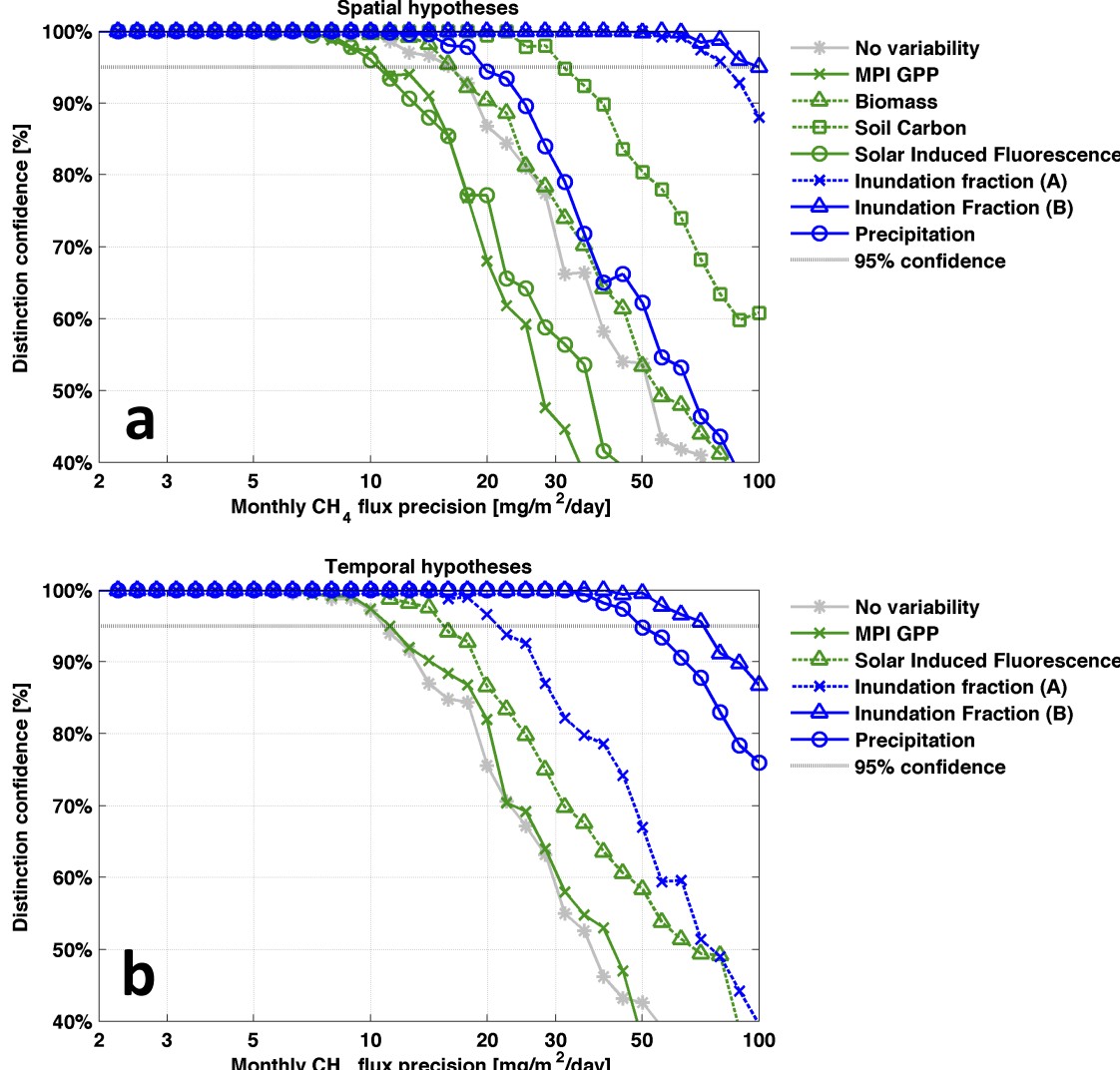

**Figure 4:** Distinction confidence between Amazon basin spatial and temporal wetland CH$_4$ emission hypotheses against monthly ~333km x 333km CH$_4$ flux precision. Spatial and temporal wetland CH$_4$ emission hypotheses are distinguishable

with a 95% confidence at a <10 mg m$^{-2}$ day$^{-1}$ precision. For this study we define our ~333km x 333km CH$_4$ flux precision requirement as 10 mg m$^{-2}$ day$^{-1}$.

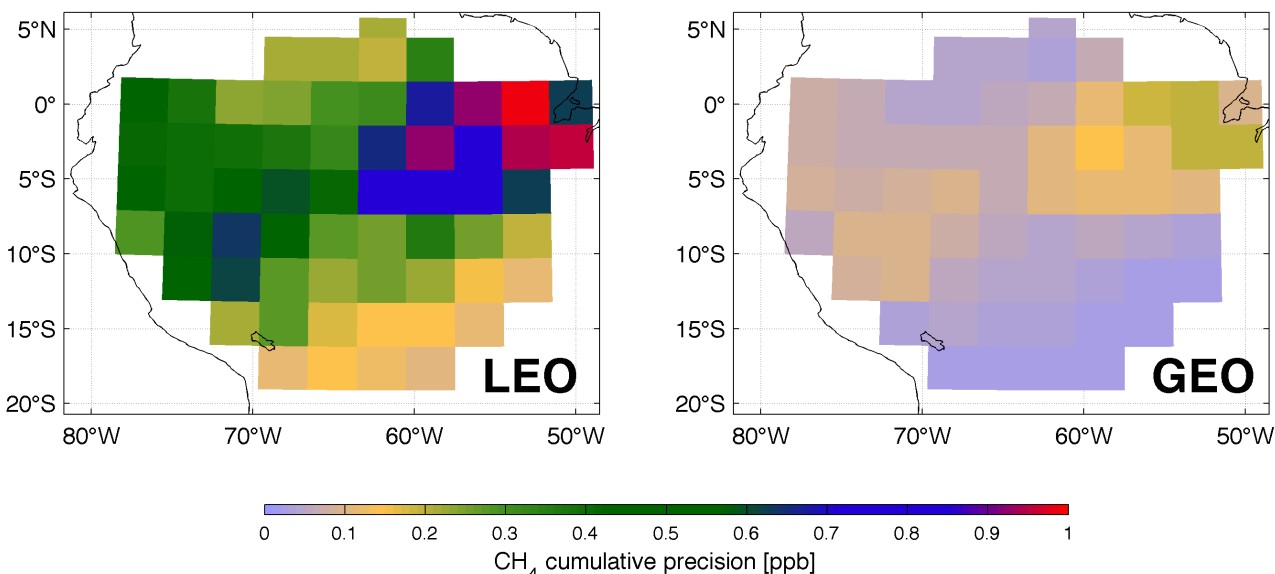

**Figure 5:** Retrieved monthly ~333km CH$_4$ cumulative precision (i.e. the combined precision of monthly-averaged CH$_4$ measurements) for LEO and GEO observing systems (OS); the OS configurations are described in Table 1.

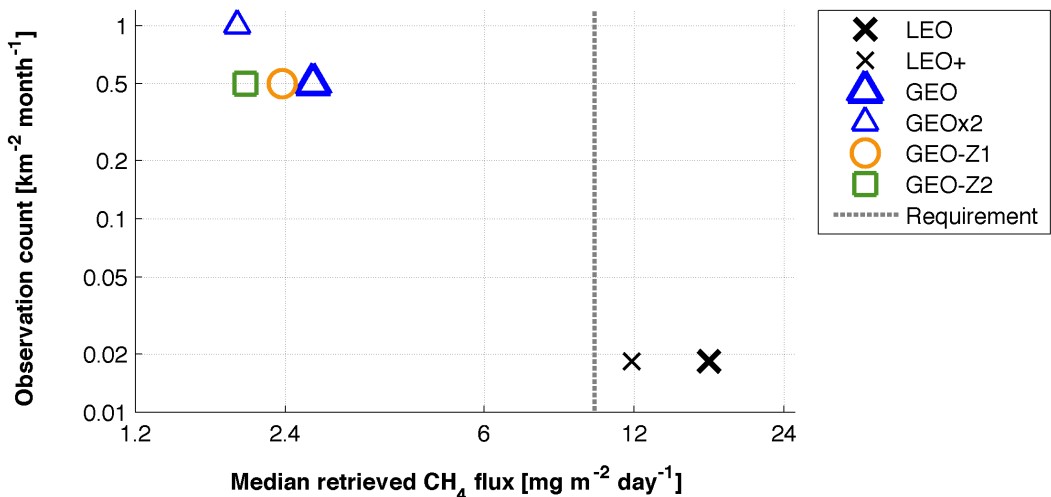

**Figure 6:** $CH_4$ observations density (observations per unit area; y-axis) versus retrievable ~333km flux precision (x-axis) for six $CH_4$ observation systems (see Table 1 for details). The "observation density" includes all attempted $CH_4$ measurements, including accepted (cloud-free) and rejected (cloudy) observations.

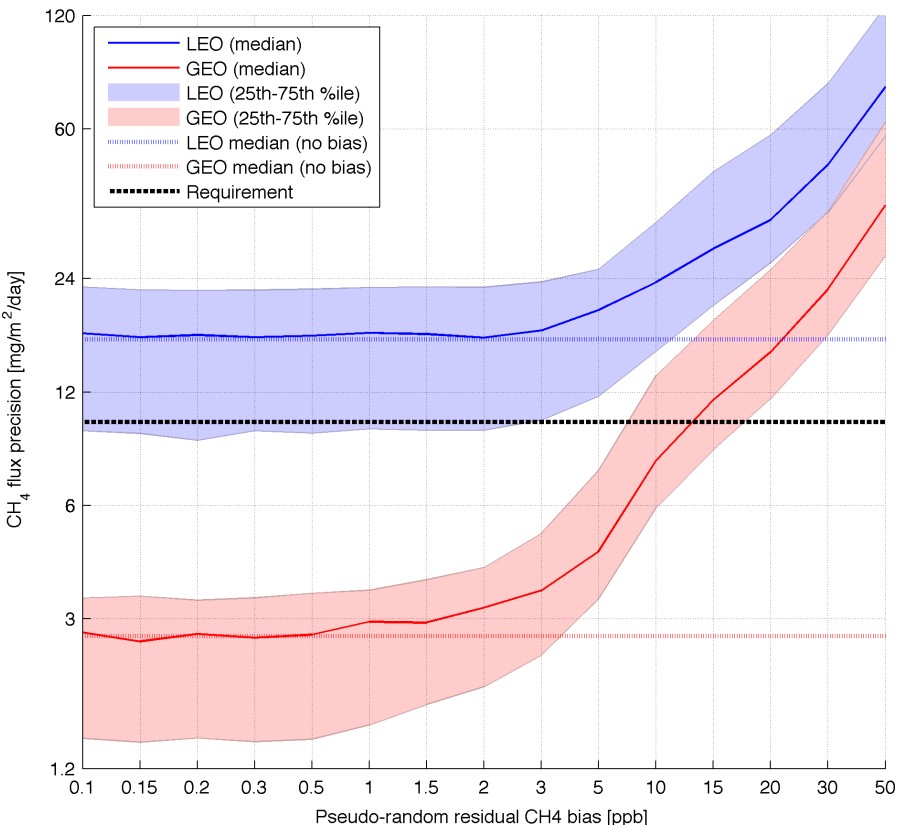

**Figure 7:** Retrieved GEO and LEO flux precision for L = ~333km with modelled pseudo-random residual bias error. See table 1 for details on GEO and LEO CH$_4$ observing systems.

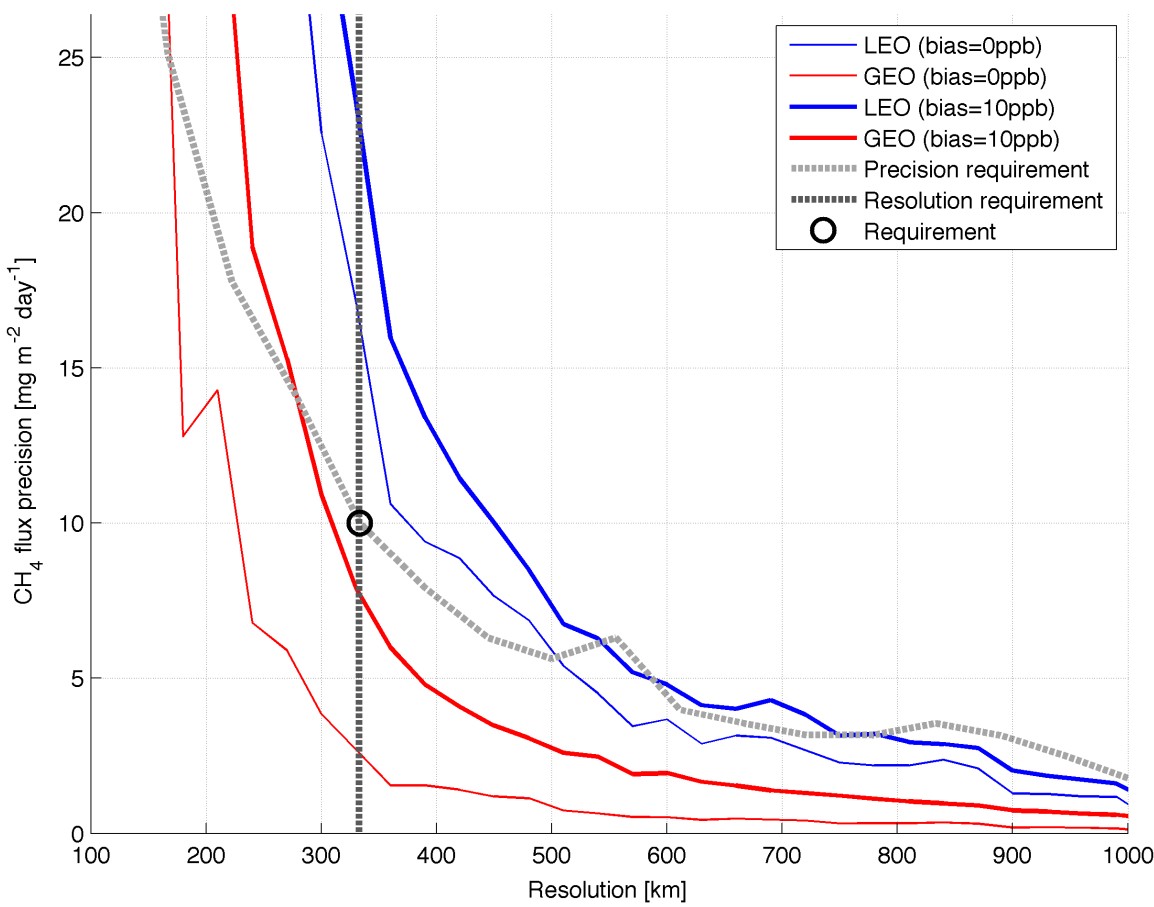

**Figure 8:** Median retrieved LEO and GEO CH$_4$ fluxes for $L = 150 – 990$km; the dashed lines indicate precision and resolution requirements. See table 1 for details on GEO and LEO CH$_4$ observing systems. The bias value of 10ppb indicates modelled systematic CH$_4$ measurement biases with 100km spatial correlations (see section 2.3).

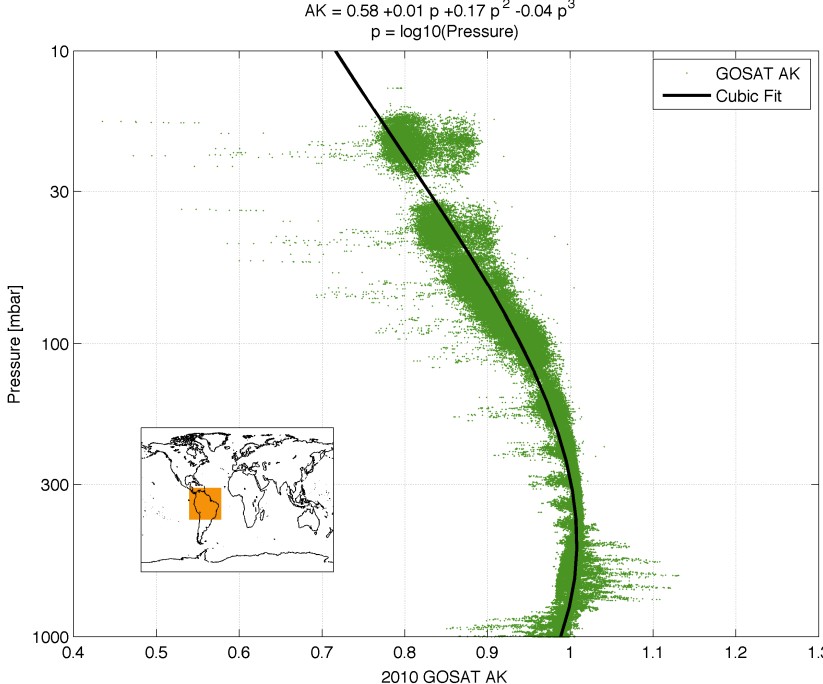

**Figure A1.** January to December 2010 GOSAT averaging kernels (AK) for the broader Amazon region (green dots). The black line denotes the AK cubic fit (w.r.t. pressure $p$; equation shown at the top of the figure). This AK was used to vertically weight the LPDM footprint and sample WRF $CH_4$ concentrations (see Appendix D).

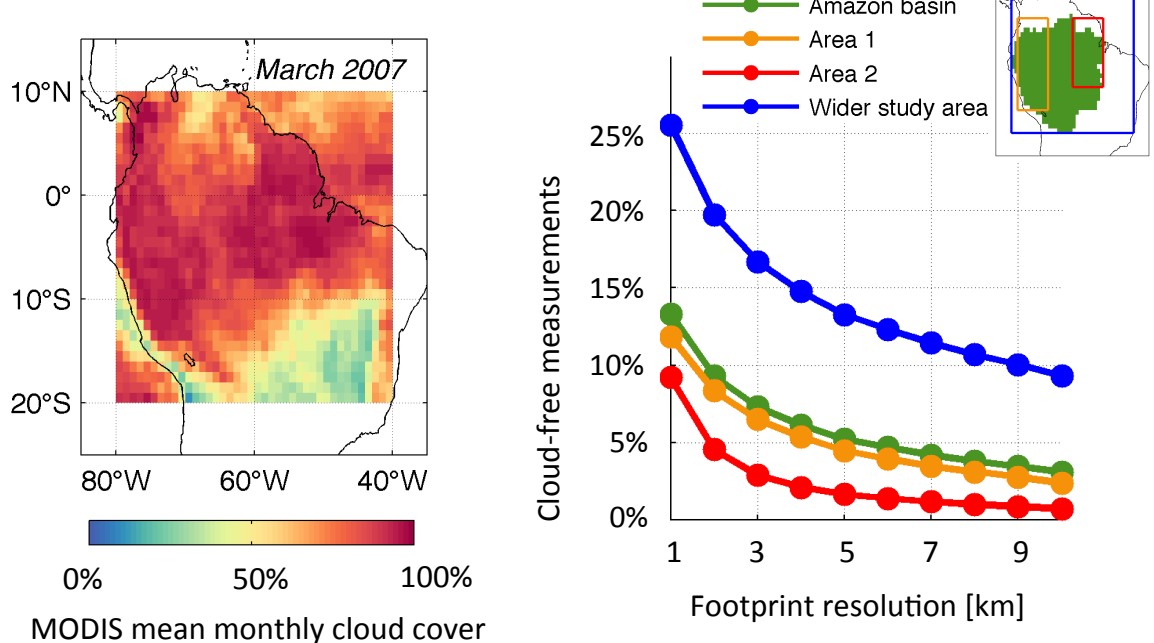

Figure B1. **Left:** March 2007 mean MODIS cloud cover aggregated to 1° ×1°. **Right:** Summary of March 2007 cloud-free observations versus footprint size for the broader study area, the Amazon river basin, and two sub-regions (east and west Amazon river basin).

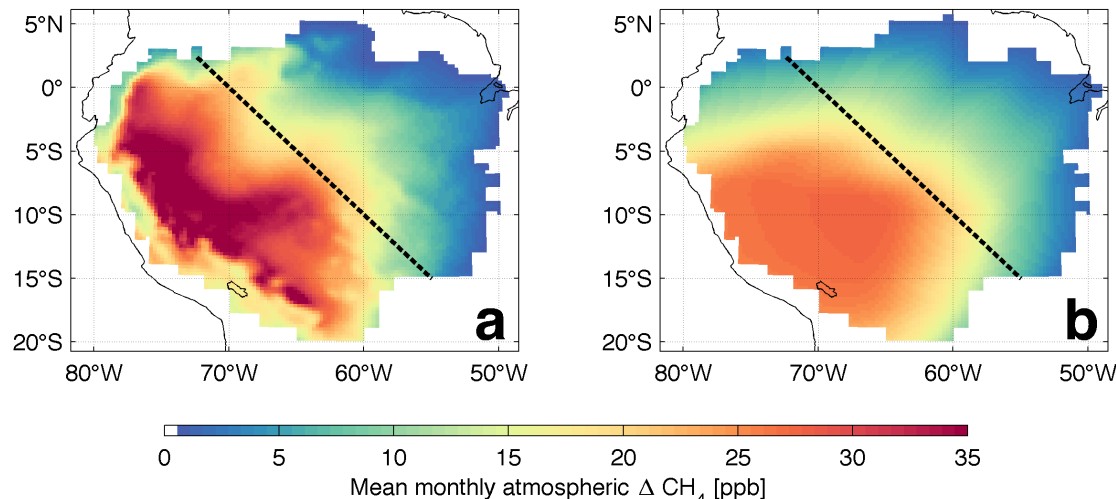

**Figure D1.** March 2007 simulations of atmospheric CH$_4$ concentration enhancements – based on 12 mg m$^{-2}$ day$^{-1}$ fluxes throughout the Amazon basin – derived using the WRF atmospheric transport model (**a**) and the LPDM influence function approach (**b**). The dashed line denotes our delineation of "North-East Amazon basin" and "South-West Amazon basin" regions (see Appendix C).