# Peer review of "What are the greenhouse gas observing system requirements for reducing fundamental biogeochemical process uncertainty? Amazon wetland CH4 emissions as a case study."

_Atmospheric Chemistry and Physics, 2016_

## Referee Comment (RC1) · Anonymous Referee #1 · 20 May 2016

The paper by Bloom et al. investigates the required performance parameters of satellite missions aimed at gaining quantitative insight into the biogeochemical processes driving methane wetland emission in the Amazon region. To this end, the authors first examine the variability (in space, time, magnitude) of the carbon cycle and hydrological processes that control CH$_4$ emissions. Then, they use observing system experiments to derive mission requirements (spatial and temporal resolution; precision) that would allow for disentangling the processes under natural variability. The study covers satellite concepts in low-earth-orbit (LEO) as well as in geostationary orbit (GEO).

[Figure]

The applied methodology is most interesting since it outlines an approach how to quantitatively derive mission requirements based on the actual variability of the targeted process parameters. I would tend to criticize the study as being too simplistic in one or the other way outlined below. But certainly, the paper is well written, methods are robust and rigorous, and thus, it is suitable for publication in ACP after considering my questions/comments below.

Questions/comments:

(1) A shortcoming of the study is the assumption of purely random error sources implying that measurement uncertainty improves with the square root of the number of binned soundings. This assumption results in maps such as Figure 4 where the measurement precision of GEO soundings binned on $300 \times 300$ km$^2$ is in the range of 0.1 ppb (given 1800 ppb background) which is a clearly unrealistic assumption for the overall measurement error. Experience with the current generation of passive greenhouse gas sounders such as GOSAT and OCO-2 tells that, at aggregated scales, random errors are dwarfed by systematic errors which typically exceed 0.1 ppb by far. Systematic errors are hard to address and, indeed, the manuscript concedes the neglect of systematic errors but a major caveat should be issued when discussing the achievable flux precisions.

(2) The manuscript restricts the advantage of a GEO sounder to massively enhanced data density. Wouldn't it make sense to actually exploit the quasi-contiguous temporal sampling of a GEO sounder? A GEO sounder would allow for resolving variability due to source and transport patterns on the time scale of hours. Running an inverse model with monthly flux resolution (and probably imposed sub-monthly variability) might simply discard some of the available process information.

Technical comments:

P5,L16: Focusing the study on March reduces data amount and related logistics but it neglects seasonal variability. Is there any indication that March is a benign or malign

case? For example: is the $CH_4$ flux precision requirement of 3 mg $CH_4/m^2/day$ valid for all seasons?

P5,L16: MODIS cannot provide information on diurnal variability in cloud cover. Would you expect a significant effect e.g. for choosing an optimal LEO overpass or for optimizing GEO revisits?

P7,L8: Looking at the correlation matrix (Figure A1), there is substantial correlation among (C1, C2, C4) and (H1,H2) on spatial scales down to 100 km which means that they would be hard to distinguish by an observing system. So, actually, the requirement L≤300 km only allows for discriminating carbon and hydrological controls but not for discriminating the type of carbon (except for C3 vs (C1,C2,C4)) or the type hydrological process (except for H3 vs (H1,H2)). Is that correct? Probably, this should be discussed in more detail.

P9,L1: It would be appropriate to cite an original TROPOMI paper at least once (instead of Wecht et al., 2014, repeatedly): P. Veefkind, I. Aben, K. McMullan, H. Förster, J. de Vries, G. Otter, J. Claas, H.J. Eskes, J.F. de Haan, Q. Kleipool, M. van Weele, O. Hasekamp, R. Hoogeveen, J. Landgraf, R. Snel, P. Tol, P. Ingmann, R. Voors, B. Kruizinga, R. Vink, H. Visser, P.F. Levelt, TROPOMI on the ESA Sentinel-5 Precursor: A GMES mission for global observations of the atmospheric composition for climate, air quality and ozone layer applications, Remote Sensing of Environment, Volume 120, 15 May 2012, Pages 70-83, ISSN 0034-4257, http://dx.doi.org/10.1016/j.rse.2011.09.027.

P10,L9: "March and September 2007". The rest of the paper is restricted to March. So, I guess, September needs to be removed.

Equation (2): The multiplication of the vectors $\mathbf{N}$ and $\mathbf{O}$ is not a scalar product but an element-wise multiplication, right? Probably, this needs to be stated somewhere.

P11,L7: Is the unperturbed $CH_4$ flux assumed constant (12 mg/$m^2$/day) throughout the domain?

P11,L17: Figure A2 -> Figure C1

P11,L11: A further advantage of GEO is several revisits per day.

Appendices: It would be useful to have a meaningful title for the appendices (instead of only Appenix A, B, C).

Equation C1: What is the inverse of a vector, $\mathbf{f}'^{-1}$?

P16,L22: Figure A1 -> Figure A2.

---

## Referee Comment (RC2) · Anonymous Referee #2 · 26 May 2016

This study presents an OSSE for different hypothetical LEO and GEO satellite instruments. The focus is on the requirements on these observing systems for obtaining process-relevant information on wetland emissions in the Amazon region. As explained below some assumptions are made, which are not well justified but have a potentially large influence on the conclusions. These will have to be dealt with in a satisfactory manner to make this paper suitable for publication in ACP.

GENERAL COMMENTS

Autocorrelation scales have been derived for several parameters to motivate the choice

of spatial scale that the measurements should be able to resolve in order for the OS to help us gain process understanding. It is presented as a novel approach that could be applied to other related problems. Although I appreciate the attempt to derive such scales (which indeed addresses an important question), I do not agree that the presented method solves this problem. The reason is that the results presented in figure 3 depend on the scale of the data sets that are used. What is shown is the autocorrelation of parameters that are averaged on a scale of 0.5x0.5 degree. If the resolution of the datasets were much higher, then other more local processes would contribute to variability shortening the overall auto-correlation scale. Indeed it is questionable whether the methane emission from a local pond really correlates with one that is 100 km away. What is the motivation to use datasets at 0.5x0.5 degree? If the processes themselves motivate this choice then this should be explained. In absence of such a motivation it is a probably more a practical choice. I have no problem with this choice as long as its limitation is made clear, and that it requires reconsideration for any other application.

If it is considered important that the inversion resolves the autocorrelation scale then it is not sufficient to evaluate the posterior uncertainty at that scale. This is because the off-diagonals of the posterior covariance matrix might indicate that neighboring fluxes are not independently determined. In this study, however, the performance criterion only considers values on the diagonal. In addition, the choice of 25% confuses monthly and annual fluxes. The requirement is on monthly fluxes, but it is derived from an estimate of Melack et al on the annual time scale.

It is unclear why a special effort is made to derive requirements on horizontal resolution looking at the drivers of processes, whereas this is not done for the requirements on flux precision and temporal resolution. Since the inversion solves for net fluxes it remains unclear anyway if these requirements really allow us to constrain specific processes. Wouldn't it have been more logical to vary process model parameters to determine what is needed to resolve them? You might wonder whether it is even realistic to constrain processes only by measuring XCH4 using a single instrument. Atmospheric measurements are useful for constraining regional emission budgets, which - in combination with other information - can be used to derive improved process understanding. The OSSE approach that is taken disqualifies instruments that provide useful constraints on larger scales as part of a multi-component global monitoring system.

This OSSE is extremely (and unrealistically I would say) optimistic about the uncertainty reduction that can be achieved by averaging large numbers of data. It is mentioned that the 'cumulative' uncertainty of GEO OS may be as low as 0.02 ppb. It is probably a main reason why the GEO measurement concept performs so well in this study. In reality, however, systematic uncertainties will kick in at much reduced precisions preventing any further improvements upon averaging. Some attempt should be made to assess the sensitivity of the conclusion that improved process-understanding calls for the GEO approach, to the presence of systematic errors in the data.

Further effort is needed to quantify the impact of errors due to the simplified treatment of atmospheric transport. In general, surface fluxes are proportional to spatio-temporal concentration gradients in the atmosphere. Looking at figure C1 it becomes clear that the east-west gradient in WRF is substantially stronger than in LPDM. It has probably to do with the north- and southward transport along the Andes in WRF, which is missing in LPDM. The impact of this should be quantified.

It should be made clearer why the analysis is limited to the month of March. Many things are different in other months (atmospheric dynamics, cloud cover, CH4 fluxes, etc.). March doesn't sound like a particularly good choice as average, or representative month.

SPECIFIC COMMENTS

Page 7, line 14: 'Throughout ... CH4 emissions" I don't see why the fact that 25% is in between the dynamic ranges of monthly GPP and inundation variability would make it suitable for separating their influences. Apart from this, what justifies the assumed

linearity between these drivers and methane emissions?

Page 9, line 11: 'i.e. all accepted ... 100% cloud-free' According to Appendix B, MODIS data that is probably cloud-free are considered as fully cloud-free. These two statements do not fit together.

Page 11, equation 3: Why is $c_{L,0}$ calculated? In the end all that matters is the spread in 'c' due to the random perturbation and how it maps on 'f' using 'A'. The uncertainty in 'f' does not depend on the mean of 'c'.

Page 13, line 21: 'If Amazon CH4 fluxes .... likely be lower' This depends on the distribution of cloud cover. The wettest regions will likely be measured the least frequent. This calls for further motivation of why uniform emissions have been assumed.

Page 15, line 7: Why is the purpose of the parentheses here? Please clarify further at what p-level the autocorrelations are required to be significant, and how this is determined. For example in the following sentence if is not clear what $r_i$ refers to. Please revise the description to explain more clearly what was done.

Figure 1: What are the different lines in the inset figure?

Figure 4: Why do you call this 'cumulative precision'? Isn't it rather the precision of a 300x300km2 average?

Figure 5: Why isn't cloud filtering affecting the number of data, comparing GEO, GEO-Z1, GEO-Z2?

Figure B1: I assume that both panels represent March 2007. If so, then this should be made clear.

Figure C1: Do these values represent the total column? If so, then mention this.

Appendix B, line 18: f(omega,i) is not used in equation 1. Where do the 30x30km2 areas come from?

Appendix C, line 17: The mean in CH4 is not the relevant quantity to compare LPDM and WRF (it is the gradient in the wind direction that matters).

TECHNICAL CORRECTIONS

-
* * *

---

## Author Comment (AC1) · 28 Oct 2016

**We thank the reviewers for their constructive feedback and suggested corrections. Below we have addressed each individual comment from reviewers 1 and 2 (reviewer comments are shown in italics; our responses to the reviewer comments are shown in bold). In light of the reviewer comments, our revised manuscript now includes (a) an analysis on the role of CH$_4$ retrieval systematic biases, and (b) a more robust quantification of the CH$_4$ flux requirements. We believe that the following revisions have substantially improved the overall quality**

[Figure]

**of our manuscript.**
* * *
*Anonymous Referee 1*

*The paper by Bloom et al. investigates the required performance parameters of satellite missions aimed at gaining quantitative insight into the biogeochemical processes driving methane wetland emission in the Amazon region. To this end, the authors first examine the variability (in space, time, magnitude) of the carbon cycle and hydrological processes that control $CH_4$ emissions. Then, they use observing system experiments to derive mission requirements (spatial and temporal resolution; precision) that would allow for disentangling the processes under natural variability. The study covers satellite concepts in low-earth-orbit (LEO) as well as in geostationary orbit (GEO).*

*The applied methodology is most interesting since it outlines an approach how to quantitatively derive mission requirements based on the actual variability of the targeted process parameters. I would tend to criticize the study as being too simplistic in one or the other way outlined below. But certainly, the paper is well written, methods are robust and rigorous, and thus, it is suitable for publication in ACP after considering my questions/comments below.*

*Questions/comments:*

**(1.1)** *(1) A shortcoming of the study is the assumption of purely random error sources implying that measurement uncertainty improves with the square root of the number of binned soundings. This assumption results in maps such as Figure 4 where the measurement precision of GEO soundings binned on 300×300 km2 is in the range of 0.1 ppb (given 1800 ppb background) which is a clearly unrealistic assumption for the overall measurement error. Experience with the current generation of passive greenhouse gas sounders such as GOSAT and OCO-2 tells that, at aggregated scales, random errors are dwarfed by systematic errors which typically exceed 0.1 ppb by far. Systematic errors are hard to address and, indeed, the manuscript concedes the neglect of systematic errors but a major caveat should be issued when discussing the achievable*

*flux precisions.*

**We have now amended our analysis with an explicit simulation of $CH_4$ residual bias errors. We describe the incorporation of a residual $CH_4$ bias structure at the end of section 2.3 of the revised manuscript. We now show that the relative advantage of a GEO mission - in comparison to a LEO mission - decreases with increasing $CH_4$ bias (Figure 7).**

**(1.2)***(2) The manuscript restricts the advantage of a GEO sounder to massively enhanced data density. Wouldn't it make sense to actually exploit the quasi-contiguous temporal sampling of a GEO sounder? A GEO sounder would allow for resolving variability due to source and transport patterns on the time scale of hours. Running an inverse model with monthly flux resolution (and probably imposed sub-monthly variability) might simply discard some of the available process information.*

**We agree that additional constraints may be achievable under certain process scenarios; for example, emissions from spatially concentrated wetland $CH_4$ sources (e.g. across the main stem of the Amazon river) could potentially be constrained based on higher resolution $CH_4$ concentration gradients, and fluxes can be estimated using alternative approaches. Conversely, monthly $CH_4$ inversions are more suitable for spatially and temporally diffuse $CH_4$ emission process scenarios. We now discuss the additional potential advantages of GEO OS in the revised manuscript.**

*Technical comments:*

**(1.3)** *P5,L16: Focusing the study on March reduces data amount and related logistics but it neglects seasonal variability. Is there any indication that March is a benign or malign case? For example: is the $CH_4$ flux precision requirement of 3 mg $CH_4$/m2/day valid for all seasons?*

**We have now included a more robust quantification of the $CH_4$ precision require-**

ment: for a given resolution requirement, we derive a year-round precision re-
quirement (now 10 mg m$^{-2}$ day$^{-1}$) as the CH$_4$ precision needed to statistically
distinguish between wetland CH$_4$ process hypotheses with a 95% confidence.

In the reviewer's words, March 2007 is a "malign case": we now state that "the
atmospheric CH$_4$ OS requirement as the ability to meet the CH$_4$ flux resolution
and precision requirements during the cloudiest time of year". We also clarify
that March 2007 is the cloudiest month in the Jan - Apr 2007 season (84% cloud
cover) and it is considerably higher than the subsequent dry season (46% - 56%
cloud cover).

*(1.4) P5,L16: MODIS cannot provide information on diurnal variability in cloud cover.
Would you expect a significant effect e.g. for choosing an optimal LEO overpass or for
optimizing GEO revisits?*

We agree with the reviewer that diurnal variability may amount to a key compo-
nent of assessing and optimizing GEO and LEO missions. Based on ERA-interim
cloud-cover re-analyses, we show that the annual mean diurnal coefficient of
variation of cloud-free Amazon basin spans 7% - 80% (median = 29%). Given
the non-linear relationship between data yield and 1km x 1km cloud-free domain
shown in Figure B1, we highlight that choice of diurnal variability could have a
substantial influence on LEO and GEO data yield. We now make these points in
the discussion section of our revised manuscript.

*(1.5) P7,L8: Looking at the correlation matrix (Figure A1), there is substantial correla-
tion among (C1, C2, C4) and (H1,H2) on spatial scales down to 100 km which means
that they would be hard to distinguish by an observing system. So, actually, the re-
quirement L ≤ 300 km only allows for discriminating carbon and hydrological controls
but not for discriminating the type of carbon (except for C3 vs (C1,C2,C4)) or the type
hydrological process (except for H3 vs (H1,H2)). Is that correct? Probably, this should
be discussed in more detail.*

**We have now removed this figure from the revised manuscript, since our precision derivation approach implicitly accounts for both spatial and temporal correlations (see response to 1.3).**

**(1.6)** *P9,L1: It would be appropriate to cite an original TROPOMI paper at least once (instead of Wecht et al., 2014, repeatedly): P. Veefkind, I. Aben, K. McMullan, H. Forster, J. de Vries, G. Otter, J. Claas, H.J. Eskes, J.F. de Haan, Q. Kleipool, M. van Weele, O. Hasekamp, R. Hoogeveen, J. Landgraf, R. Snel, P. Tol, P. Ingmann, R. Voors, B. Kruizinga, R. Vink, H. Visser, P.F. Levelt, TROPOMI on the ESA Sentinel-5 Precursor: A GMES mission for global observations of the atmospheric composition for climate, air quality and ozone layer applications, Remote Sensing of Environment, Volume 120, 15 May 2012, Pages 70-83, ISSN 0034-4257, http://dx.doi.org/10.1016/j.rse.2011.09.027.*

**We now cite the Veefkind et al., (2012) paper as a reference for the TROPOMI mission.**

**(1.7)** *P10,L9: "March and September 2007". The rest of the paper is restricted to March. So, I guess, September needs to be removed.*

**We have now removed "September".**

**(1.8)** *Equation (2): The multiplication of the vectors N and O is not a scalar product but an element-wise multiplication, right? Probably, this needs to be stated somewhere.*

**We now use an appropriate symbol and explicitly state this in the text.**

**(1.9)** *P11,L7: Is the unperturbed $CH_4$ flux assumed constant (12 mg/m2/day) throughout the domain?*

**In response to the second reviewer's comments (see responses to 2.2 and 2.9) we now report flux uncertainties in mg m$^{-2}$ day$^{-1}$, and we have revised equation 5 accordingly. Since the explicit definition of f{L,0} is now redundant, it has been removed from the revised manuscript.**

**(1.10)** *P11,L17: Figure A2 -> Figure C1*

**We now correctly reference this figure.**

**(1.11)** *P11,L11: A further advantage of GEO is several revisits per day.*

**We now clearly state this in the revised manuscript**

**(1.12)** *Appendices: It would be useful to have a meaningful title for the appendices (instead of only Appenix A, B, C).*

**We have now added descriptive titles to Appendices A-D.**

**(1.13)** *Equation C1: What is the inverse of a vector, f'1?*

**We have now added a sentence to better clarify that f' is an N x N array, comprised of N flux vectors.**

**(1.14)** *P16,L22: Figure A1 -> Figure A2.*

**Figure reference now corrected**

*Anonymous Referee 2*

*This study presents an OSSE for different hypothetical LEO and GEO satellite instruments. The focus is on the requirements on these observing systems for obtaining process-relevant information on wetland emissions in the Amazon region. As explained below some assumptions are made, which are not well justified but have a potentially large influence on the conclusions. These will have to be dealt with in a satisfactory manner to make this paper suitable for publication in ACP.*

*GENERAL COMMENTS*

**(2.1)** *Autocorrelation scales have been derived for several parameters to motivate the choice of spatial scale that the measurements should be able to resolve in order for the OS to help us gain process understanding. It is presented as a novel approach that could be applied to other related problems. Although I appreciate the attempt to de-*

*rive such scales (which indeed addresses an important question), I do not agree that the presented method solves this problem. The reason is that the results presented in figure 3 depend on the scale of the data sets that are used. What is shown is the autocorrelation of parameters that are averaged on a scale of 0.5x0.5 degree. If the resolution of the datasets were much higher, then other more local processes would contribute to variability shortening the overall auto-correlation scale. Indeed it is questionable whether the methane emission from a local pond really correlates with one that is 100 km away. What is the motivation to use datasets at 0.5x0.5 degree? If the processes themselves motivate this choice then this should be explained. In absence of such a motivation it is a probably more a practical choice. I have no problem with this choice as long as its limitation is made clear, and that it requires reconsideration for any other application.*

**We agree with the reviewer that our assessment of carbon and hydrological process variable correlation scales requires reconsideration for any subsequent application. We now clarify that the auto-correlation scales are specific to Amazon river basin; we also highlight the limitation of our auto-correlation approach, and we clarify that finer-scale analyses may require higher resolution datasets to quantify GHG measurement requirements.**

**We also agree with the reviewer that finer-scale variability from higher-resolution datasets could potentially contribute to alternative assessments of auto-correlation scales. However, in our derivation of Moran's I at each L, we aggregate our data at an L x L resolution (see Appendix A), and therefore fine-scale variability is averaged out (regardless of the native resolution of the dataset).**

**(2.2)** *If it is considered important that the inversion resolves the autocorrelation scale then it is not sufficient to evaluate the posterior uncertainty at that scale. This is because the off-diagonals of the posterior covariance matrix might indicate that neighboring fluxes are not independently determined. In this study, however, the performance criterion only considers values on the diagonal. In addition, the choice of 25% confuses*

*monthly and annual fluxes. The requirement is on monthly fluxes, but it is derived from an estimate of Melack et al on the annual time scale.*

**We agree with the reviewer that using a "%" precision is misleading. We now present flux precision in CH$_4$ flux units (mg m$^{-2}$ day$^{-1}$) throughout the manuscript; the units are now consistent with our revised precision requirement (10 mg m$^{-2}$ day$^{-1}$; see response to reviewer comment 1.3).**

**We agree with the reviewer that "off-diagonal" error correlations in retrieved fluxes would likely indicate that neighbouring fluxes are not independently determined. However, as long as all diagonal terms meet the precision requirement (10 mg m$^{-2}$ day$^{-1}$), the OS can resolve underlying spatial flux patterns at the required precision (regardless of posterior error covariance).**

**(2.3)** *It is unclear why a special effort is made to derive requirements on horizontal resolution looking at the drivers of processes, whereas this is not done for the requirements on flux precision and temporal resolution. Since the inversion solves for net fluxes it remains unclear anyway if these requirements really allow us to constrain specific processes. Wouldn't it have been more logical to vary process model parameters to determine what is needed to resolve them? You might wonder whether it is even realistic to constrain processes only by measuring XCH$_4$ using a single instrument. Atmospheric measurements are useful for constraining regional emission budgets, which - in combination with other information - can be used to derive improved process understanding. The OSSE approach that is taken disqualifies instruments that provide useful constraints on larger scales as part of a multi-component global monitoring system.*

**We now include a quantification of the CH$_4$ flux precision requirements for distinguishing between both spatial and temporal CH$_4$ emission hypotheses (see response to reviewer comment 1.3). We have also now included a lagged Pearson's correlation analysis to determine the temporal process control correlation lengths.**

We agree with the reviewer that varying process parameters in a model is potentially a useful approach for quantifying the OS needed to improve process understanding. However, due to the scarcity of top-down constraints and in-situ measurements in tropical wetland environments, little is known about whether current models are able to capture the first-order spatial and temporal variability of wetlands. We discuss and state that model approaches can be used - albeit with due caution - to define $CH_4$ OS measurement requirements in of the revised manuscript.

Finally, we now highlight the need to investigate the added advantages of a multi-component global monitoring system in the revised manuscript.

 **(2.4)** *This OSSE is extremely (and unrealistically I would say) optimistic about the uncertainty reduction that can be achieved by averaging large numbers of data. It is mentioned that the 'cumulative' uncertainty of GEO OS may be as low as 0.02 ppb. It is probably a main reason why the GEO measurement concept performs so well in this study. In reality, however, systematic uncertainties will kick in at much reduced precisions preventing any further improvements upon averaging. Some attempt should be made to assess the sensitivity of the conclusion that improved process-understanding calls for the GEO approach, to the presence of systematic errors in the data.*

We agree with the reviewer that systematic biases are a limiting factor in the potential performance of a GEO approach. We have now included a residual $CH_4$ bias analysis to address this comment (see response to comment 1.1).

 **(2.5)** *Further effort is needed to quantify the impact of errors due to the simplified treatment of atmospheric transport. In general, surface fluxes are proportional to spatio-temporal concentration gradients in the atmosphere. Looking at figure C1 it becomes clear that the east-west gradient in WRF is substantially stronger than in LPDM. It has probably to do with the north- and southward transport along the Andes in WRF, which is missing in LPDM. The impact of this should be quantified.*

We agree with the reviewer that the LPDM approach underestimates the east-west gradient (see response to 2.18), and we now highlight that the LPDM provides a conservative estimate on the observable $CH_4$ gradients across the region. To quantify the potential bias stemming from underestimated $CH_4$ gradient across the Amazon domain, we conduct a sensitivity test on the GEO and LEO median flux precision retrievals, where the LPDM-derived transport operator is multiplied by 1.5. We find that this leads to an inversely proportional (33%) reduction in the GEO and LEO flux precision (the sensitivity test results are reported in the revised manuscript).

**(2.6)** *It should be made clearer why the analysis is limited to the month of March. Many things are different in other months (atmospheric dynamics, cloud cover, $CH_4$ fluxes, etc.). March doesn't sound like a particularly good choice as average, or representative month.*

We now define our OS requirements as the ability to resolve monthly $CH_4$ fluxes at the required resolution and precision during the cloudiest part of the 2007 wet season (see response to comment 1.3). We also highlight that March is the cloudiest month in the 2007 wet season. Finally, we highlight the need to investigate the role seasonal transport variability (amongst other factors) on GEO and LEO $CH_4$ flux retrievals.

*SPECIFIC COMMENTS*

**(2.7)** *Page 7, line 14: 11Throughout ... $CH_4$ emissions" I don't see why the fact that 25% is in between the dynamic ranges of monthly GPP and inundation variability would make it suitable for separating their influences. Apart from this, what justifies the assumed linearity between these drivers and methane emissions?*

We have now addressed this concern with a more robust derivation of $CH_4$ flux requirements (see response to comment 1.3).

[Figure]

**(2.8)** *Page 9, line 11: 'i.e. all accepted ... 100% cloud-free' According to Appendix B, MODIS data that is probably cloud-free are considered as fully cloud-free. These two statements do not fit together.*

**We have grouped "probably cloud free" and "cloud free" flags together, and "probably cloudy" and "cloudy" flags together. We have clarified this in the revised manuscript, and we have added a sentence in the appendix to clarify our assumption.**

**(2.9)** *Page 11, equation 3: Why is $c_{L,0}$ calculated? In the end all that matters is the spread in 'c' due to the random perturbation and how it maps on 'f' using 'A'. The uncertainty in 'f' does not depend on the mean of 'c'.*

**We agree with the reviewer's statement, since our derivation of f (equation 5) is independent of $c_{L,0}$. For the sake of simplicity, we now set all $c_{L,0}$ values to zero.**

**(2.10)** *Page 13, line 21: 'If Amazon $CH_4$ fluxes .... likely be lower' This depends on the distribution of cloud cover. The wettest regions will likely be measured the least frequent. This calls for further motivation of why uniform emissions have been assumed.*

**In the revised manuscript, we now clearly define our OS requirement as the ability to statistically distinguish between biogeochemical process hypotheses based on cloud cover statistics during the cloudiest time of the 2007 wet season (see response to comment 1.3).**

**(2.11)** *Page 15, line 7: Why is the purpose of the parentheses here? Please clarify further at what p-level the autocorrelations are required to be significant, and how this is determined. For example in the following sentence if is not clear what $r_i$ refers to. Please revise the description to explain more clearly what was done.*

**We have now revised this sentence to better convey our derivation of the Moran's**

**I p-value.**

**(2.12)** *Figure 1: What are the different lines in the inset figure?*

**The green lines denote the average WETCHIMP model Amazon basin monthly CH$_4$ emissions. We have revised the figure caption to clarify this.**

**(2.13)** *Figure 4: Why do you call this 'cumulative precision'? Isn't it rather the precision of a 300x300km2 average?*

**We now explicitly define CH$_4$ "cumulative precision" in the revised manuscript. For the sake of clarity, we also define CH$_4$ "cumulative precision" in the figure caption.**

**(2.14)** *Figure 5: Why isn't cloud filtering affecting the number of data, comparing GEO, GEO- Z1, GEO-Z2?*

**Observations per unit area include all attempted measurements (both cloud and cloud-free measurements). We have revised the figure caption to reflect this.**

**(2.15)** *Figure B1: I assume that both panels represent March 2007. If so, then this should be made clear.*

**Figure caption updated**

**(2.16)** *Figure C1: Do these values represent the total column? If so, then mention this.*

**Figure caption updated**

**(2.17)** *Appendix B, line 18: f(omega,i) is not used in equation 1. Where do the 30x30km2 areas come from?*

**We now correctly use 'phi' (as opposed to 'f') in referencing the fraction of cloud-free observations in equation 1. We have also corrected '30 × 30km' to 'L × L'.**

**(2.18)** *Appendix C, line 17: The mean in CH$_4$ is not the relevant quantity to compare*

*LPDM and WRF (it is the gradient in the wind direction that matters).*

**We now also report the LPDM-approach and WRF gradients across the domain in Appendix D (13.14ppb and 17.24ppb respectively); we calculate the gradients as the CH$_4$ difference between the North-East and South-West sub-regions of Amazon basin domain. We have also updated the LPDM-WRF figure to mark the delineation between the "North-East" and "South-West" regions.**

---

## Author Response (AR1)

**We thank the reviewers for their constructive feedback and suggested corrections. Below we have addressed each individual comment from reviewers #1 and #2 (reviewer comments are shown in italics; our responses to the reviewers' comments are shown in bold). In light of the reviewers' comments, our revised manuscript now includes (a) an analysis on the role of CH4 retrieval systematic biases, and (b) a more robust quantification of the CH₄ flux requirements. All manuscript changes are highlighted as 'tracked changes' in the revised manuscript (the bracketed line numbers denote the corresponding line numbers in the revised manuscript). We believe that the following revisions have substantially improved the overall quality of our manuscript.**

**Anonymous Referee #1**

*The paper by Bloom et al. investigates the required performance parameters of satellite missions aimed at gaining quantitative insight into the biogeochemical processes driving methane wetland emission in the Amazon region. To this end, the authors first examine the variability (in space, time, magnitude) of the carbon cycle and hydrological processes that control CH4 emissions. Then, they use observing system experiments to derive mission requirements (spatial and temporal resolution; precision) that would allow for disentangling the processes under natural variability. The study covers satellite concepts in low-earth-orbit (LEO) as well as in geostationary orbit (GEO).*

*The applied methodology is most interesting since it outlines an approach how to quantitatively derive mission requirements based on the actual variability of the targeted process parameters. I would tend to criticize the study as being too simplistic in one or the other way outlined below. But certainly, the paper is well written, methods are robust and rigorous, and thus, it is suitable for publication in ACP after considering my questions/comments below.*

*Questions/comments:*

*(1.1) (1) A shortcoming of the study is the assumption of purely random error sources implying that measurement uncertainty improves with the square root of the number of binned soundings. This assumption results in maps such as Figure 4 where the measurement precision of GEO soundings binned on 300×300 km2 is in the range of 0.1 ppb (given 1800 ppb background) which is a clearly unrealistic assumption for the overall measurement error. Experience with the current generation of passive greenhouse gas sounders such as GOSAT and OCO-2 tells that, at aggregated scales, random errors are dwarfed by systematic errors which typically exceed 0.1 ppb by far. Systematic errors are hard to address and, indeed, the manuscript concedes the neglect of systematic errors but a major caveat should be issued when discussing the achievable flux precisions.*

We have now amended our analysis with an explicit simulation of $CH_4$ residual bias errors. We describe the incorporation of a residual $CH_4$ bias structure at the end of section 2.3 of the revised manuscript (P14 L3-17). We now show that the relative advantage of a GEO mission – in comparison to a LEO mission – decreases with increasing $CH_4$ bias (Figure 7). We have revised the results and discussion section (P15 L17-24), and the abstract (P1 L25 – P2 L3) to reflect this.

*(1.2) (2) The manuscript restricts the advantage of a GEO sounder to massively enhanced data density. Wouldn't it make sense to actually exploit the quasi-contiguous temporal sampling of a GEO sounder? A GEO sounder would allow for resolving variability due to source and transport patterns on the time scale of hours. Running an inverse model with monthly flux resolution (and probably imposed sub-monthly variability) might simply discard some of the available process information.*

We agree that additional constraints may be achievable under certain process scenarios; for example, emissions from spatially concentrated wetland $CH_4$ sources (e.g. across the main stem of the Amazon river) could potentially be constrained based on higher resolution $CH_4$ concentration gradients, and fluxes can be estimated using alternative approaches. Conversely, monthly $CH_4$ inversions are more suitable for spatially and temporally diffuse $CH_4$ emission process scenarios. We now discuss the additional potential advantages of GEO OS in the revised manuscript (P17 L13-16).

Technical comments:

*(1.3) P5,L16: Focusing the study on March reduces data amount and related logistics but it neglects seasonal variability. Is there any indication that March is a benign or malign case? For example: is the $CH_4$ flux precision requirement of 3 mg CH4/m2/day valid for all seasons?*

We have now included a more robust quantification of the $CH_4$ precision requirement: for a given resolution requirement, we derive a year-round precision requirement (now 10mg m$^{-2}$ day$^{-1}$) as the $CH_4$ precision needed to statistically distinguish between wetland $CH_4$ process hypotheses with a 95% confidence. The precision requirement and its derivation are described in P7 L4-6, P8 L15 –P9 L6 and Appendix B. The results of our precision requirement analysis are shown in Figure 4.

In the reviewer's words, March 2007 is a "malign case": we now state that "the atmospheric $CH_4$ OS requirement as the ability to meet the $CH_4$ flux resolution and precision requirements during the cloudiest time of year" (P5 L19-20). We also clarify March 2007 is the cloudiest month in the Jan – Apr 2007 season (84% cloud cover) and it is considerably higher than the subsequent dry season (46% - 56% cloud cover) in P5 L22-24.

*(1.4) P5,L16: MODIS cannot provide information on diurnal variability in cloud cover. Would you expect a significant effect e.g. for choosing an optimal LEO overpass or for optimizing GEO revisits?*

**We agree with the reviewer that diurnal variability may amount to a key component of assessing and optimizing GEO and LEO missions. Based on ERA-interim cloud-cover re-analyses, we show that the annual mean diurnal coefficient of variation of cloud-free Amazon basin spans 7% - 80% (median = 29%). Given the non-linear relationship between data yield and 1km x 1km cloud-free domain shown in Figure B1, we highlight that choice of diurnal variability could have a substantial influence on LEO and GEO data yield. We now make these points in the discussion section of our revised manuscript (P16 L22 – P17 L2).**

*(1.5) P7,L8: Looking at the correlation matrix (Figure A1), there is substantial correlation among (C1, C2, C4) and (H1,H2) on spatial scales down to 100 km which means that they would be hard to distinguish by an observing system. So, actually, the requirement L≤300 km only allows for discriminating carbon and hydrological controls but not for discriminating the type of carbon (except for C3 vs (C1,C2,C4)) or the type hydrological process (except for H3 vs (H1,H2)). Is that correct? Probably, this should be discussed in more detail.*

**We have now removed this figure (formerly "Figure A1") from the revised manuscript, since our precision derivation approach implicitly accounts for both spatial and temporal correlations (see response to 1.3 and Figure 4).**

*(1.6) P9,L1: It would be appropriate to cite an original TROPOMI paper at least once (instead of Wecht et al., 2014, repeatedly): P. Veefkind, I. Aben, K. McMullan, H. Förster, J. de Vries, G. Otter, J. Claas, H.J. Eskes, J.F. de Haan, Q. Kleipool, M. van Weele, O. Hasekamp, R. Hoogeveen, J. Landgraf, R. Snel, P. Tol, P. Ingmann, R. Voors, B. Kruizinga, R. Vink, H. Visser, P.F. Levelt, TROPOMI on the ESA Sentinel-5 Precursor: A GMES mission for global observations of the atmospheric composition for climate, air quality and ozone layer applications, Remote Sensing of Environment, Volume 120, 15 May 2012, Pages 70-83, ISSN 0034-4257, http://dx.doi.org/10.1016/j.rse.2011.09.027.*

**We now cite the Veefkind et al., (2012) paper as a reference for the TROPOMI mission in P4 L23 and P15 L14.**

*(1.7) P10,L9: "March and September 2007". The rest of the paper is restricted to March. So, I guess, September needs to be removed.*

**We have now removed "September".**

*(1.8) Equation (2): The multiplication of the vectors N and O is not a scalar product but an element-wise multiplication, right? Probably, this needs to be stated somewhere.*

**We now use an appropriate symbol and explicitly state this in P12 L21.**

*(1.9) P11,L7: Is the unperturbed CH4 flux assumed constant (12 mg/m2/day) throughout the domain?*

**In response to the second reviewer's comments (see responses to 2.2 and 2.9) we now report flux uncertainties in mg m$^{-2}$ day$^{-1}$, and we have revised equation 5 accordingly. Since the explicit definition of f{L,0} is now redundant, it has been removed from the revised manuscript.**

*(1.10) P11,L17: Figure A2 -> Figure C1*

**We now correctly reference figure "D1" (previously figure C1) in P13 L11 of the revised manuscript.**

*(1.11) P11,L11: A further advantage of GEO is several revisits per day.*

**We now clearly state this in P14 L24 – P15 L2 of the revised manuscript**

*(1.12) Appendices: It would be useful to have a meaningful title for the appendices (instead of only Appenix A, B, C).*

**We have now added descriptive titles to Appendices A-D.**

*(1.13) Equation C1: What is the inverse of a vector, f′–1?*

**We have now added a sentence to better clarify that f' is an N × N array, comprised of N flux vectors (P21 L16-18).**

*(1.14) P16,L22: Figure A1 -> Figure A2.*

**Figure reference now corrected**

*Anonymous Referee #2*

This study presents an OSSE for different hypothetical LEO and GEO satellite instruments. The focus is on the requirements on these observing systems for obtaining process-relevant information on wetland emissions in the Amazon region. As explained below some assumptions are made, which are not well justified but have a potentially large influence on the conclusions. These will have to be dealt with in a satisfactory manner to make this paper suitable for publication in ACP.

*GENERAL COMMENTS*

*(2.1) Autocorrelation scales have been derived for several parameters to motivate the choice of spatial scale that the measurements should be able to resolve in order for the OS to help us gain process understanding. It is presented as a novel approach that could be applied to other related problems. Although I appreciate the attempt to derive such scales (which indeed addresses an important question), I do not agree that the presented method solves this problem. The reason is that the results presented in figure 3 depend on the scale of the data sets that are used. What is shown is the autocorrelation of parameters that are averaged on a scale of 0.5x0.5 degree. If the resolution of the datasets were much higher, then other more local processes would contribute to variability shortening the overall auto-correlation scale. Indeed it is questionable whether the methane emission from a local pond really correlates with one that is 100 km away. What is the motivation to use datasets at 0.5x0.5 degree? If the processes themselves motivate this choice then this should be explained. In absence of such a motivation it is a probably more a practical choice. I have no problem with this choice as long as its limitation is made clear, and that it requires reconsideration for any other application.*

**We agree with the reviewer that our assessment of carbon and hydrological process variable correlation scales requires reconsideration for any subsequent application. We now clarify that the auto-correlation scales are specific to the Amazon river basin; we also highlight the limitation of our auto-correlation approach, and we clarify that finer-scale analyses may require higher resolution datasets to quantify GHG measurement requirements (P17 L4 – 5 and P17 L8-11).**

**We also agree with the reviewer that finer-scale variability from higher-resolution datasets could potentially contribute to alternative assessments of auto-correlation scales. However, in our derivation of Moran's I at each *L*, we aggregate our data at an L × L resolution (see Appendix A), and therefore fine-scale variability is averaged out (regardless of the native resolution of the dataset).**

*(2.2) If it is considered important that the inversion resolves the autocorrelation scale then it is not sufficient to evaluate the posterior uncertainty at that scale. This is because the off-diagonals of the posterior covariance matrix might indicate that neighboring fluxes are not independently determined. In this study, however, the performance criterion only considers values on the diagonal. In addition, the choice of 25% confuses monthly and annual fluxes. The requirement is on monthly fluxes, but it is derived from an estimate of Melack et al on the annual time scale.*

**We agree with the reviewer that using a "%" precision is misleading. We now present flux precision in flux units (mg CH$_4$ m$^{-2}$ day$^{-1}$) throughout the manuscript and in Figures (6-8). The units are now consistent with our**

**revised precision requirement (10mg/m²/day; see response to reviewer comment 1.3).**

**We agree with the reviewer that "off-diagonal" error correlations in retrieved fluxes would likely indicate that neighbouring fluxes are not independently determined. However, as long as all diagonal terms meet the precision requirement (10mg/m²/day), the OS can resolve underlying spatial flux patterns at the required precision (regardless of posterior error covariance).**

*(2.3) It is unclear why a special effort is made to derive requirements on horizontal resolution looking at the drivers of processes, whereas this is not done for the requirements on flux precision and temporal resolution. Since the inversion solves for net fluxes it remains unclear anyway if these requirements really allow us to constrain specific processes. Wouldn't it have been more logical to vary process model parameters to determine what is needed to resolve them? You might wonder whether it is even realistic to constrain processes only by measuring XCH4 using a single instrument. Atmospheric measurements are useful for constraining regional emission budgets, which - in combination with other information - can be used to derive improved process under- standing. The OSSE approach that is taken disqualifies instruments that provide useful constraints on larger scales as part of a multi-component global monitoring system.*

**We now include a quantification of the $CH_4$ flux precision requirements for distinguishing between both spatial and temporal $CH_4$ emission hypotheses (see response to reviewer comment 1.3). We have also now included a lagged Pearson's correlation analysis to determine the temporal process control correlation lengths (P8 L5-11).**

**We agree with the reviewer that varying process parameters in a model is potentially a useful approach for quantifying the OS needed to improve process understanding. However, due to the scarcity of top-down constraints and in-situ measurements in tropical wetland environments (P6 L23 – P7 L1), little is known about whether current models are able to capture the first-order spatial and temporal variability of wetlands. We have expanded our discussion in P17 L20 - P18 L9 to clearly state that model approaches can be used – albeit with due caution – to define $CH_4$ OS measurement requirements of the revised manuscript.**

**We also highlight the need to investigate the added advantages of a multi-component global monitoring system in P17 L16-18 of the revised manuscript.**

*(2.4) This OSSE is extremely (and unrealistically I would say) optimistic about the uncertainty reduction that can be achieved by averaging large numbers of data. It is mentioned that the 'cumulative' uncertainty of GEO OS may be as low as 0.02 ppb. It is probably a main reason why the GEO measurement concept performs so well in this study. In reality, however, systematic uncertainties will kick in at much reduced*

*precisions preventing any further improvements upon averaging. Some attempt should be made to assess the sensitivity of the conclusion that improved process-understanding calls for the GEO approach, to the presence of systematic errors in the data.*

**We agree with the reviewer that systematic biases are a limiting factor in the potential performance of a GEO approach. We have now included a residual CH$_4$ bias analysis to address this comment (see response to comment 1.1).**

*(2.5) Further effort is needed to quantify the impact of errors due to the simplified treatment of atmospheric transport. In general, surface fluxes are proportional to spatio-temporal concentration gradients in the atmosphere. Looking at figure C1 it becomes clear that the east-west gradient in WRF is substantially stronger than in LPDM. It has probably to do with the north- and southward transport along the Andes in WRF, which is missing in LPDM. The impact of this should be quantified.*

**We agree with the reviewer that the LPDM approach underestimates the east-west gradient (see response to 2.18), and we now highlight that the LPDM provides a conservative estimate on the observable CH$_4$ gradients across the region (P16 L12-14). To quantify the potential bias stemming from underestimated CH$_4$ gradient across the Amazon domain, we conduct a sensitivity test on the GEO and LEO median flux precision retrievals, where the LPDM-derived transport operator is multiplied by 1.5. We find that this leads to an inversely proportional (~33%) reduction in the GEO and LEO flux precision (we report this in lines P16 L14-16 of the revised manuscript).**

*(2.6) It should be made clearer why the analysis is limited to the month of March. Many things are different in other months (atmospheric dynamics, cloud cover, CH4 fluxes, etc.). March doesn't sound like a particularly good choice as average, or representative month.*

**We now define our OS requirements as the ability to resolve monthly CH$_4$ fluxes at the required resolution and precision during the cloudiest part of the 2007 wet season (see response to comment 1.3). We also highlight that March is the cloudiest month in the 2007 wet season (P5 L23-25). Finally, we highlight the need to investigate the role seasonal transport variability (amongst other factors) on GEO and LEO CH$_4$ flux retrievals (P16 L18-20).**

*SPECIFIC COMMENTS*

*(2.7)Page 7, line 14: 'Throughout ... CH4 emissions" I don't see why the fact that 25% is in between the dynamic ranges of monthly GPP and inundation variability would make it suitable for separating their influences. Apart from this, what justifies the assumed linearity between these drivers and methane emissions?*

We have now addressed this concern with a more robust derivation of $CH_4$ flux requirements (see response to comment 1.3).

*(2.8)Page 9, line 11: 'i.e. all accepted ... 100% cloud-free' According to Appendix B, MODIS data that is probably cloud-free are considered as fully cloud-free. These two statements do not fit together.*

We have grouped "probably cloud free" and "cloud free" flags together, and "probably cloudy" and "cloudy" flags together. We have clarified this in P11 L8-10 in the revised manuscript, and we have added a sentence in Appendix C to clarify our assumptions (P21 L4-6).

*(2.9)Page 11, equation 3: Why is c{L,0} calculated? In the end all that matters is the spread in 'c' due to the random perturbation and how it maps on 'f' using 'A'. The uncertainty in 'f' does not depend on the mean of 'c'.*

We agree with the reviewer's statement, since our derivation of $f$ (equation 5) is independent of c{L,0}. For the sake of simplicity, we now set all c{L,0} values to zero: we clarify this in P13 L15-16 of the revised manuscript.

*(2.10)Page 13, line 21: 'If Amazon CH4 fluxes .... likely be lower' This depends on the distribution of cloud cover. The wettest regions will likely be measured the least frequent. This calls for further motivation of why uniform emissions have been assumed.*

In the revised manuscript, we now clearly define our OS requirement as the ability to statistically distinguish between biogeochemical process hypotheses based on cloud cover statistics during the cloudiest time of the 2007 wet season (P5 L23-25, and see response to comment 1.3).

*(2.11)Page 15, line 7: Why is the purpose of the parentheses here? Please clarify further at what p-level the autocorrelations are required to be significant, and how this is determined. For example in the following sentence if is not clear what r_i refers to. Please revise the description to explain more clearly what was done.*

We have now revised this sentence to better convey our derivation of the Moran's I p-value (P19 L6-8).

*(2.12) Figure 1: What are the different lines in the inset figure?*

The green lines denote the average WETCHIMP model Amazon basin monthly $CH_4$ emissions. We have revised the figure caption to clarify this.

*(2.13)Figure 4: Why do you call this 'cumulative precision'? Isn't it rather the precision of a 300x300km2 average?*

We now explicitly define $CH_4$ "cumulative precision" in P10 L10-11 of the revised manuscript. For the sake of clarity, we also define $CH_4$ "cumulative precision" in the figure caption (now Figure 5).

*(2.14)Figure 5: Why isn't cloud filtering affecting the number of data, comparing GEO, GEO- Z1, GEO-Z2?*

Observations per unit area include all attempted measurements (both cloud and cloud-free measurements). We have revised the figure caption (now Figure 6) to reflect this.

*(2.15)Figure B1: I assume that both panels represent March 2007. If so, then this should be made clear.*

Figure caption updated

*(2.16) Figure C1: Do these values represent the total column? If so, then mention this.*

Figure caption updated

*(2.17)Appendix B, line 18: f(omega,i) is not used in equation 1. Where do the 30x30km2 areas come from?*

We now correctly use 'phi' (as opposed to 'f') in referencing the fraction of cloud-free observations in equation 1. We have also corrected '30x30km' to 'L × L'. (P21 L8-9)

*(2.18) Appendix C, line 17: The mean in CH4 is not the relevant quantity to compare LPDM and WRF (it is the gradient in the wind direction that matters).*

We now also report the LPDM-approach and WRF gradients across the domain in Appendix D (P23 L11-13; 13.14ppb and 17.24ppb respectively); we calculate the gradients as the $CH_4$ difference between the North-East and South-West sub-regions of Amazon basin domain. We have also updated Figure D1 to mark the delineation between the "North-East" and "South-West" regions.

*Additional changes*

We have rectified a minor bug in our Moran's I code. We have updated the results and Figure 3 accordingly.

For consistency with the new precision requirement derivation, we have changed our spatial $CH_4$ requirement from "300km" to "~333km".

We have updated Figure 8 to include bias simulations and the precision and resolution requirements.

We have removed the first paragraph of Appendix A, as the text was redundant.

[revised manuscript text omitted]